# Towards the Training of Deeper Predictive Coding Neural Networks

## Abstract

Predictive coding networks are neural models that perform inference through an iterative energy minimization process, whose operations are local in space and time. While effective in shallow architectures, they suffer significant performance degradation beyond five to seven layers. In this work, we show that this degradation is caused by exponentially imbalanced errors between layers during weight updates, and by predictions from the previous layers not being effective in guiding updates in deeper layers. Furthermore, when training models with skip connections, the energy propagated by the residuals reaches higher layers faster than that propagated by the main pathway, affecting test accuracy. We address the first issue by introducing a novel precision-weighted optimization of latent variables that balances error distributions during the relaxation phase, the second issue by proposing a novel weight update mechanism that reduces error accumulation in deeper layers, and the third one by using auxiliary neurons that slow down the propagation of the energy in the residual connections. Empirically, our methods achieve performance comparable to backpropagation on deep models such as ResNets, opening new possibilities for predictive coding in complex tasks.

## 1 Introduction

Training deep learning models is extremely expensive in terms of energy consumption. To address this problem, a recent direction of research is studying the use of alternative accelerators that leverage the properties of physical systems to perform computations (Wright et al., 2022; Momeni et al., 2024), such as in-memory computations using memristor crossbars (Tsai et al., 2018; Haensch et al., 2018). However, transitioning to new hardware without altering the main algorithm — error backpropagation (Rumelhart et al., 1986) — has proven to be challenging due to two central issues: the requirement for sequential forward and backward passes, and the need to analytically compute gradients of a global cost function. These issues do not arise when using learning algorithms that rely on computations that are local in space and time (i.e., all updates rely on locally available information only) (Bengio, 2014; Hinton, 2022). A popular example is equilibrium propagation, a framework that allows for the learning of the parameters of a neural network by simulating a physical system brought to an equilibrium (Scellier & Bengio, 2017). This physical system is usually defined via an energy function that describes the state of a neural network in terms of its weights and neurons, with different functions describing different systems (Krotov & Hopfield, 2016).

In recent years, researchers have devoted significant effort to scaling up the deployment of energy-based learning algorithms. Two recent works have benchmarked multiple variants of common learning algorithms using the Hopfield energy function (Scellier et al., 2024) and the predictive coding energy (Pinchetti et al., 2024), showing that this class of models can match the performance of standard deep learning when training relatively shallow networks of up to five or seven layers. However, this success does not extend to deeper architectures, where performance degrades substantially. Notably, the degradation is even more severe in predictive coding implementations of residual networks, which perform worse than equally deep models without skip connections. Since the major advances of modern deep learning rely on very deep architectures with residual connections, understanding and overcoming these limitations is a critical step toward scaling predictive coding networks to the regimes where deep learning has been most successful.

To understand the poor scalability of predictive coding networks, it is instructive to examine how energy propagates across depth. It has recently been shown that in a three-layer model, the energy concentrated in a layer can be up to an order of magnitude larger than the energy concentrated in the layer before (Pinchetti et al., 2024). While such shallow models can still achieve good test accuracies, we conjecture that this 'energy imbalance' becomes a critical bottleneck in deeper architectures, leading to performance degradation in a way that is conceptually similar to the *vanishing gradient* problem (Hochreiter, 1998). More precisely, this imbalance prevents the effective propagation of energy — and crucially, the associated error information — from the output layer back to the early layers, creating two problems: first, it prevents the model from fully leveraging its depth, as early layers receive insufficient error signals for effective training; second, latent states may diverge substantially from their forward pass values due to the excessive energy in later layers.

Addressing this energy imbalance requires mechanisms that can adaptively modulate the relative influence of different layers as errors propagate through the network, a challenge similar to what biological neural systems solve through precision-based regulation. In predictive processing theories, precision refers to the estimated reliability of a prediction error (formally, its inverse variance), and it is thought to be dynamically regulated to balance bottom-up and top-down signals across cortical hierarchies (Feldman & Friston, 2010; Bastos et al., 2012). Despite its central role in biological inference, most machine learning formulations of predictive coding set precision to 1 for simplicity (Whittington & Bogacz, 2017), thereby overlooking its potential as a powerful mechanism to stabilize learning dynamics. The first contribution of this work is to propose leveraging precision weighting to regularize energy propagation in predictive coding networks.

We begin by analyzing the energy propagation of deep convolutional architectures for both predictive coding (PC) and incremental PC (iPC) — a recently introduced variant of PC that updates weights and neurons simultaneously Salvatori et al. (2024). Building on the insights from this analysis, we propose time-dependent precisions that address the identified issues. Our results show that this substantially improves test accuracy, supporting our conjecture of a causal link between energy propagation and empirical performance. More broadly, we propose two algorithmic improvements (spiking precision and a novel weight-update mechanism) and two structural improvements (PC-tailored batch normalization and auxiliary neurons for skip connections) that enable PC to achieve competitive performance with backprop on image classification benchmarks, including VGG models with up to 15 layers and ResNets18 on Tiny ImageNet. Our contributions can be detailed as follows:

- We show that in models trained with PC, the energy is orders of magnitude larger in layers closer to the output, supporting the hypothesis that information fails to propagate effectively to the earlier layers. This phenomenon is less pronounced in iPC, where continuous weight updates rapidly reduce the excess energy. However, these updates result in even lower test accuracy.

- To mitigate this imbalance, we propose dynamical precision-weightings that depend on both time and layer depth. The most effective variant, which we call *spiking precision*, applies very large precisions as soon as the energy reaches a given layer, thereby boosting it forward. Experiments show that this method regulates the energy imbalance and improves test accuracy in deep PC models. In the case of iPC, spiking precisions alone are already sufficient to achieve performance comparable to backpropagation in deep networks.

- To further improve the performance of standard PC models, we introduce a novel weight-update mechanism that modifies how parameters are updated. This method combines predictions computed at initialization (hence adding a degree of implausibility, as they have to be stored in memory) with the neural activities at convergence, resulting in more effective updates. With this approach, PC attains performance on par with backpropagation and iPC on deep convolutional models. In addition, we propose a variant of batch normalization (Ioffe & Szegedy, 2015) tailored for PC, which further enhances performance.

- While effective for VGG-like models, when training PC-based ResNets He et al. (2016), we still observe a significant drop in performance. We conjecture that this is caused by the energy propagated by the residuals, which reach higher layers faster than that propagated by the main pathway, disrupting learning dynamics. We show that this can be addressed by adding extra families of neurons inside the skip connections, that have the sole goal of slowing down the feedback signal of the skip connections so that it reaches the higher layers at the same time as the main one. The results show that such auxiliary neurons allow models trained with PC and iPC to reach performance comparable to these of backprop on ResNet18.

## 2 RELATED WORKS

**Equilibrium Propagation (EP).**    EP is a learning algorithm for supervised learning that is largely inspired by contrastive learning on continuous Hopfield networks (Movellan, 1991). Here, neural activities are updated in two phases: In the first, to minimize an energy function defined on the parameters of the neural network; in the second, to minimize the same energy with the addition of a loss function defined on the labels (Scellier & Bengio, 2017). Interestingly, these two phases allow us to approximate the gradient of the loss function up to arbitrary levels of accuracy using finite difference coefficients (Zucchet & Sacramento, 2022). The consequence is that EP can be seen as a technique that allows the minimization of loss functions using arbitrary physical systems that can be brought to an equilibrium, and it has hence been studied in a large number of domains (Scellier, 2024; Kendall et al., 2020). The state of the art is that EP models are able to match the performance of BPTT (BP Through Time) on models with 5 and 7 hidden layers (Scellier et al., 2024), with the exception of hybrid models, which manage to achieve a good performance on models with 15 layers by alternating blocks of layers trained with BP and blocks trained with EP (Nest & Ernoult, 2024).

**Predictive Coding (PC).**    The formulation of PC that we use here was developed to model hierarchical information processing in the brain (Rao & Ballard, 1999; Friston, 2005). Intuitively, this theory states that neurons and synapses at one level of the hierarchy are updated to better predict the activities of the neurons of the layers below, and minimize the *prediction error*. Interestingly, the same algorithm can be used as a training algorithm for deep neural networks (Whittington & Bogacz, 2017), where several similarities with backpropagation were observed (Song et al., 2020; Salvatori et al., 2022). To this end, it has been used in a large number of machine learning tasks, from image generation and classification to natural language processing and associative memory (Sennesh et al., 2024; Salvatori et al., 2023; Pinchetti et al., 2022; Ororbia & Kifer, 2020; Salvatori et al., 2021). Again, the state of the art has been reached by training convolutional models with 5 hidden layers, with performance starting to get worse as soon as we use models with 7 layers (Pinchetti et al., 2024). To this end, a recent interesting work has proposed $\mu PC$, a theoretical framework that allows the training of very deep feedforward models Innocenti et al. (2025). However, this work does not tackle non-feedforward layers, and does not test on datasets larger than MNIST. The connection between PC and EP is well explained by the concept of bi-level optimization (Zucchet & Sacramento, 2022), where the neural activities used for learning are the equilibrium state of a physical system.

## 3 BACKGROUND

Let us consider a network with $L$ layers, and let us denote $\mathbf{W}^l$ and $\mathbf{x}_t^l$ the weight parameters and the neural activities of layer $l$, respectively. Note that, differently from standard models, the neural activities are variables of the model, optimized over time steps $t$. This optimization is performed with the goal of allowing the activities of every layer to predict those of the layer below. Together with the neural activities, the two other quantities related to single neurons are the *prediction* $\mu_{\mathbf{t}}^{\mathbf{l}} = \mathbf{W}^{\mathbf{l}} \mathbf{f} \left( \mathbf{x}_{\mathbf{t}}^{\mathbf{l-1}} \right)$, given by the layer-wise operation through an activation function, and the *prediction error*, defined as the deviation of the actual activity from the prediction, that is, $\varepsilon_t^l = \mathbf{x}_t^l - \mu_t^l$. A fourth quantity, usually overlooked in machine learning applications but of vital importance in neuroscience, is the *precision*, or *covariance* $\Sigma_t^l$ of a specific neuron [1]. Differently from the standard literature, we consider the covariance to be time-dependent. Furthermore, instead of updating it to minimize an objective function as done in previous works (Ofner et al., 2021), we will manually define the rule that governs its updates. The predictive coding energy is then the sum of the squared norms of the precision-weighted prediction errors of every layer over time:

$$E_t = \frac{1}{2} \sum_{l=1}^{L} \frac{\|\mathbf{x}_t^l - \mu_t^l\|^2}{\Sigma_t^l} = \frac{1}{2} \sum_{l=1}^{L} \frac{\|\varepsilon_t^l\|^2}{\Sigma_t^l}, \qquad (1)$$

where we consider covariances to be layer-dependent: all the neurons of the same layer will have the same covariance. To this end, we use the same notation when the covariance $\Sigma_t^l$ is a scalar, or a diagonal matrix whose entries are equal to such a scalar.

---

[1]In predictive coding, and more generally in statistics, the precision matrix is the matrix inverse of the covariance matrix. In this work, we follow the standard convention and divide the prediction error by a factor of $\Sigma$, instead of multiplying it by a factor of $p$.

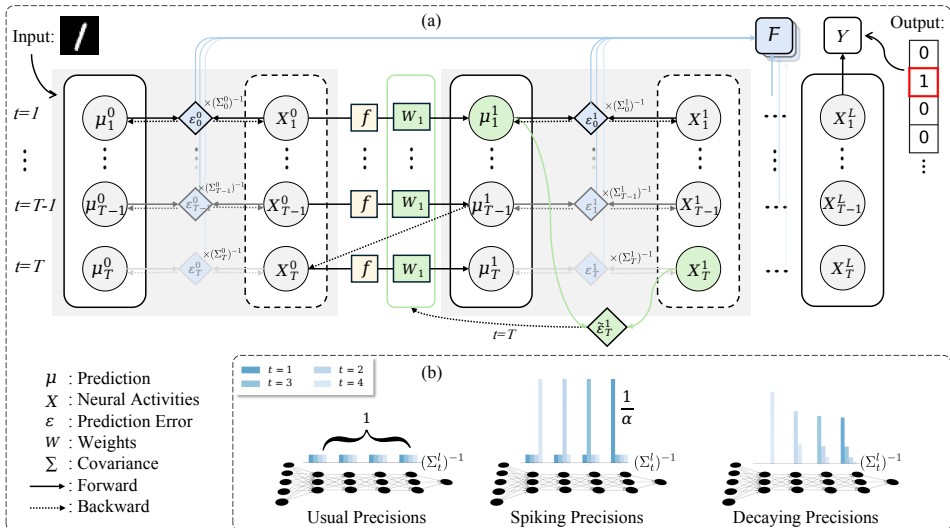

Figure 1: (a) Evolution of predictive coding models over multiple time iterations. The green diamond $\tilde{\varepsilon}_T^{l+1}$ refers to the information needed to compute the proposed forward updates. The rest of the figure represents the standard components and mechanisms of a predictive coding network. (b) Visualization of the proposed precision-weighting strategies, where the height of the bar is proportional to the precision at different time steps.

**Training.** Suppose that we are provided with a labeled data point $(\mathbf{o}, \mathbf{y})$. Training is then performed via a form of bi-level optimization (Zucchet & Sacramento, 2022), divided into three phases. In the first phase, the neural activities of every neuron are initialized via a forward pass, that is, by setting $\mathbf{x}_0^l = \mu_0^l$ for every layer $l$, with $x_0^0 = o$. In the second phase, which we call the *inference* phase, we clamp the output neurons to the label (i.e., we set $\mathbf{x}_L = \mathbf{y}$) and update the neural activities to minimize the total energy. At $t = 0$, this energy is exactly the supervised loss, as all internal errors are zero and only the output mismatch contributes. The update rule is then:

$$\Delta \mathbf{x}_t^l = -\alpha \frac{\partial E_t}{\partial \mathbf{x}_t^l} = \alpha \left( \frac{\boldsymbol{\varepsilon}_t^l}{\Sigma_t^l} - \mathbf{W}^{(l+1)\top} \frac{\boldsymbol{\varepsilon}_t^{l+1}}{\Sigma_t^l} \odot f'(\mathbf{x}_t^l) \right), \tag{2}$$

where $\alpha$ is the learning rate of the neural activities. This phase will continue until it reaches the fixed number of iterations $\mathbf{T}$ or convergence. The third phase is the *learning* phase, where the neural activities $\mathbf{x}_T^l$ are fixed, and the weight parameters are updated to decrease the energy, weighted by a learning rate $\eta$, via the following equation:

$$\Delta \mathbf{W}^l = -\eta \frac{\partial E_T}{\partial \mathbf{W}^l} = -\eta \frac{\boldsymbol{\varepsilon}_T^l f(\mathbf{x}_T^{l-1})}{\Sigma_T^l}, \tag{3}$$

**Incremental PC (iPC).** An alternative to the bi-level optimization described above is iPC Salvatori et al. (2024), that differs from the standard implementation as also the weight parameters are updated at every time step $t$ until convergence, according to Eq.3. From the variational inference perspective, this schema introduces a form of *incremental* expectation-maximization Neal & Hinton (1998), and has been argued to be more biologically plausible than standard PC, as it avoids the need for a control signal that halts the updates of the neural activities and triggers the weight updates. In practice, however, this method has been shown to perform even worse than PC when it is used to train deep models such as VGG7 and VGG9 on CIFAR10 Pinchetti et al. (2022). We will show that it is this fully automatic training method that will benefit the most from hard-coded precisions, and reach the best performance on models such as ResNet18.

**Nudging.** Instead of providing the original label $y$ to the model, it is common in the literature to slightly translate the output neurons of the system $\mathbf{x}_0^L$ in the direction of $y$. More precisely, it fixes $\mathbf{x}_t^L = \mu_0^L + \beta(\mathbf{y} - \mu_0^L)$ for every time step $t$, where $\beta$ controls the supervision strength. The sign of $\beta$ determines supervision polarity: positive for standard nudging and negative for inverse supervision. Performing a stochastic sampling from $\{\beta, -\beta\}$ across training epochs and batches is called *center nudging* (Scellier et al., 2024). In practice, PC with centered nudging has been shown to be the best performing method on deep models such as VGG7 Pinchetti et al. (2024).

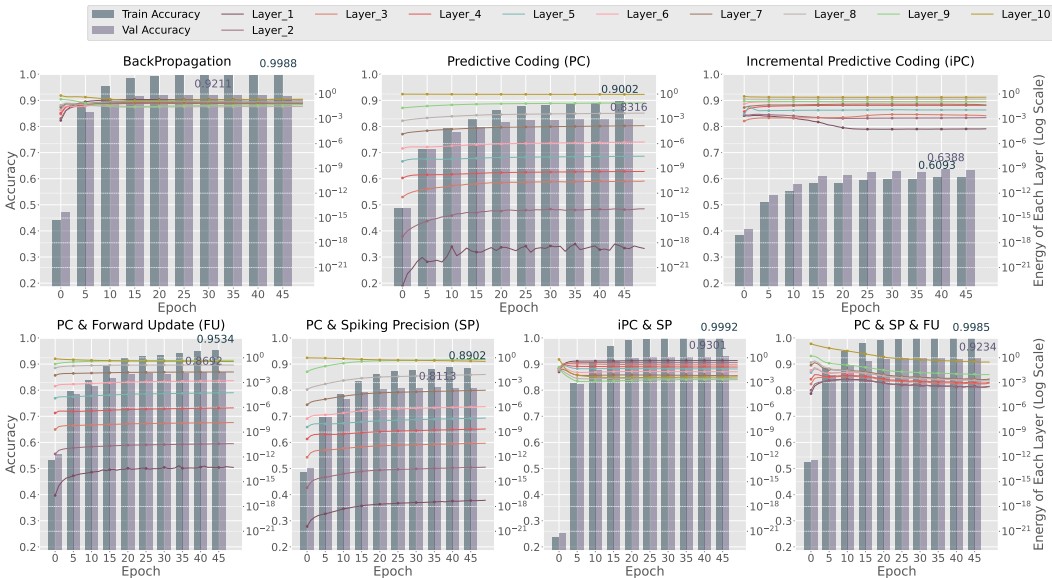

Figure 2: Normalized layer-wise energy distribution and accuracy comparison between BP and PCNs in a VGG10 on the CIFAR10 dataset. Colored curves represent the total energy of the individual layers of the model (or, the squared error of every layer for BP). The vertical lines represent the train and test accuracies of the model.

## 4 METHODS

In this section, we first study the phenomenon of energy imbalance across different layers, and then use the derived insights to propose time-dependent covariances that address it. In detail, we propose *spiking precisions*, a method that better distributes the energy across the model by dynamically updating the precisions as soon as individual neurons are reached by the energy term. In practice, we show that this largely improves the performance of models trained with PC and iPC. In the case of iPC, spiking precisions allow us to reach performance comparable to that of backprop. To further boost the performance of standard PC, we introduce a variation of the weight update rule, which leverages neural activities at initialization to perform a better update of the parameters and improve overall model performance. Figure 1(a) presents a flowchart that intuitively illustrates the modules discussed in subsequent sections, while Figure 1(b) provides a visualization of the covariance matrices.

To study the energy imbalance across different levels of the network, we have tracked the normalized total energy of each layer during training, along with the test and training accuracy, and compared it against that of BP. We have performed a broad study that tests multiple models, datasets, and setups, which we mostly report in the supplementary material, while presenting in Figure 2 the plots of the best performing models. As BP does not have a proper definition of energy, we have used the squared error of every layer computed during the backward pass, equivalent to the error of $PC$ when it comes to the update of the weight parameters.

**Results.** In models trained with PC, there is a significant energy imbalance, where early layers have up to $18$ orders of magnitude less energy than later layers, while in models trained with iPC this phenomenon is less pronounced but still present, as the layer with less energy has an energy of about $10^{-5}$. This does not happen in BP-trained models, which exhibit a more uniform energy distribution across layers. In fact, the first layer has energy above $10^{-2}$ almost the whole training. This set of experiments shows the potential reason why deep PC models do not perform well. In the bottom row, we show how combinations of our proposed methods mitigate such an energy imbalance, with the best performing ones being iPC with spiking precisions, and PC with spiking precisions and a novel update rule we will discuss later. In both cases, the first layer has an energy above the layer of lowest energy above $10^{-3}$. These two methods are also the only ones reaching test accuracy slightly better than those of backprop.

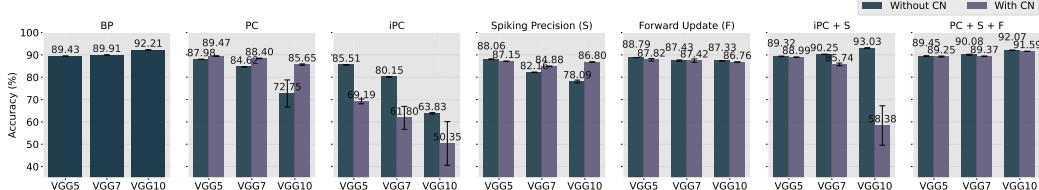

Figure 3: Test accuracies of various algorithms on the CIFAR10 dataset, evaluated on models of different depths. From the second plot onward, each pair of bars compares the performance of the algorithm with and without center nudging (CN).

## 4.1 ALGORITHMIC CONTRIBUTIONS

**Spiking Precision.** Training predictive coding models involves a critical trade-off between stability and the efficient propagation of error signals. On the one hand, large learning rates $\alpha$ for the neural activities can lead to instabilities: most of the best results in the field have been obtained using a small learning rate, such as $\alpha = 0.05$ (Pinchetti et al., 2024). On the other hand, such small learning rates can exponentially slow down the propagation of error signals across model layers, as noted in the supplementary material of (Song et al., 2020). To this end, we propose to modulate the precision having a *spike* proportional to the learning rate the first time the energy — initially concentrated in the output neurons — reaches a specific layer. In terms of temporal scheduling, this happens at $l = L - t$. For a network with $L$ layers and $T$ inference steps, the proposed spiking precision hence is:

$$\Sigma_t^l = \begin{cases} \alpha & \text{when } l = L - t, \\ 1, & \text{otherwise.} \end{cases} \tag{4}$$

Intuitively, the spikes allow the energy to be well propagated from the last to the first layer during the first $L$ iterations, with the other updates happening as usual.

**Forward Updates.** Due to large prediction errors that we find in the last layers, the neural activities observed at the end of the inference process tend to significantly deviate from their initial feed-forward values. But the feedforward values are the ones that are then used for predictions. We hence conjecture that synaptic weight updates based on $\mathbf{x}_T^l$ could potentially introduce errors that accumulate with network depth, leading to performance degradation in deeper architectures. To assess whether the proposed conjecture is correct, we introduce a new method for updating the weight parameters that uses both the starting and final states of neurons, according to the energy function defined as

$$\tilde{E}_T = \frac{1}{2} \sum_l \frac{\|\tilde{\varepsilon}_T^l\|^2}{\Sigma_t^l}, \qquad \text{where} \quad \tilde{\varepsilon}_T^l = \mathbf{x}_T^l - \mu_0^l. \tag{5}$$

Our method makes sure that weight adjustments stay connected to the initial feed-forward predictions while incorporating the refined representations obtained through iterative inference. This approach has the advantage of maintaining stability during learning and prevents the accumulation of errors in deeper layers, which is crucial for scaling PC networks, but also the disadvantage of storing information in memory, which is then used for the weight update only, making it less bio-plausible than the original formulation. A similar energy function has been used in a different way in a previous work, where the authors used it to guide the update of the neural activities instead of the weight updates (Whittington & Bogacz, 2017).

### 4.1.1 EXPERIMENTAL VALIDATION OF THE ALGORITHMIC CONTRIBUTIONS

Our hypothesis is that the proposed methods improve the performance of PC and iPC on deep models. To this end, we perform experiments on VGG-like models (Simonyan & Zisserman, 2014) — convolutional models followed by feedforward layers — on the CIFAR10 dataset, where we observe their test accuracies as a function of their depth. We again use backprop, PC, and iPC as baselines, and report the results computed with and without centered nudging in Figure 3. All the details needed to reproduce the experiments, architectures, and hyperparameters used can be found in the supplementary material.

**Results.** The barplots show that PC and iPC with and without center nudging significantly drop in test accuracy when the depth of the model is increased. In contrast, our proposed methods avoid the accuracy degradation as model depth increases. Particularly, iPC with spiking precision is the method that exhibits performance comparable to BP across VGG5/7/10 models, despite being completely local in space and time. We will later see that this is consistent with the more complex experiments, where this method is still the one achieving the best performance overall. In models without forward update, consistent with previous findings, using center nudging enhances algorithm accuracy, with more pronounced effects as the number of model layers increases in most cases. However, we found that once forward update is added, center nudging ceases to yield performance benefits.

**Linear and exponential decays.** To address the problem of energy imbalances, we used *spikes* that help propagate the energy to the first layers. However, alternative options would have been to attenuate the energy accumulation in the last layers by gradually reducing their precision over time. We therefore evaluate two variants where precisions decay either linearly or exponentially with the number of time steps, using the same setup of the experiment above, that is, a VGG10 on CIFAR10. The results, reported in

Table 1: Performance of different precision schedules on VGG10/CIFAR-10.

| Method | Test Accuracy (%) |
|---|---|
| Fixed + FU | $87.33 \pm 0.14$ |
| Lin Decay + FU | $88.35 \pm 0.12$ |
| Exp Decay + FU | $89.43 \pm 0.18$ |
| Spiking + FU | $92.07 \pm 0.10$ |

Tab. 1, show that such decays provide improvements over the baseline, confirming the benefit of dynamically modulating precisions. However, spiking precisions consistently outperform both decay schedules, underscoring that actively boosting error propagation is more effective than passively damping energy imbalances. We refer to the supplementary material for a more detailed description.

## 4.2 STRUCTURAL CONTRIBUTIONS

**Residual Connections.** Previously, we showed that our proposed methods overcome the depth limitation when training VGG-style models with 10 layers. However, this improvement does not extend to ResNet10, where performance still drops catastrophically. We conjecture that this degradation arises from a mismatch in how energy propagates through residual connections: the skip path delivers energy to higher layers earlier than the main feedforward path. Concretely, consider the residual block in Fig. 4(a): when the neurons $\mathbf{x}^{l+2}$ receive energy at time $t$, in the next step they propagate it both to $\mathbf{x}^{l+1}$ (through the feedforward connection) and to $\mathbf{x}^{l-1}$ (through the skip connection). As a result, higher-level neurons begin updating before the main stream of energy arrives.

Having two streams of energy that reach the same levels in different time steps contrasts with both the philosophy behind the spiking updates, that are supposed to boost the neural activities the first time this is reached by the energy, and iPC, that updates some weight parameters before the information that goes through the model has reached them. Furthermore, it has been shown that it makes the feedback signal diverge from the one of backprop Salvatori et al. (2022). To address this temporal mismatch, we introduce auxiliary families of neurons along each residual connection, one for every skipped layer, as shown in Fig. 4(a). These auxiliary units act as buffers that delay the propagation of energy through the skip path, ensuring that energy arriving via the residual connection reaches the start of the block synchronously with the energy propagated through the main path.

Formally, we augment the network with a total of $K$ auxiliary nodes $\mathbf{x}^{\mathbf{res_k}}$ distributed across the residual pathways. Here, $K$ is defined as the cumulative number of intermediate layers bypassed by all skip connections in the architecture. In this formulation, the prediction of every auxiliary node $\mu_t^{res_k}$ is defined by a fixed identity mapping from the preceding node, i.e., $\mu_t^{res_k} = Id(\mathbf{x_t^{res_{k-1}}})$. This structural constraint forces the feedback signal to traverse the residual path sequentially, thereby matching the propagation delay of the main pathway. The global energy $E_t$ is thus expanded to include the precision-weighted errors of these auxiliary variables:

$$E_t = \frac{1}{2} \sum_{l=1}^{L} \frac{\|\mathbf{x}_t^l - \mu_t^l\|^2}{\Sigma_t^l} + \frac{1}{2} \sum_{k=1}^{K} \frac{\|\mathbf{x}_t^{res_k} - \mu_t^{res_k}\|^2}{\Sigma_t^{res_k}}.$$

In the case of spiking updates, the covariance $\Sigma_t^{res_k}$ of an auxiliary node is set equal to the covariance $\Sigma_t^l$ of the main layer at the same hierarchical level, ensuring unified energy propagation.

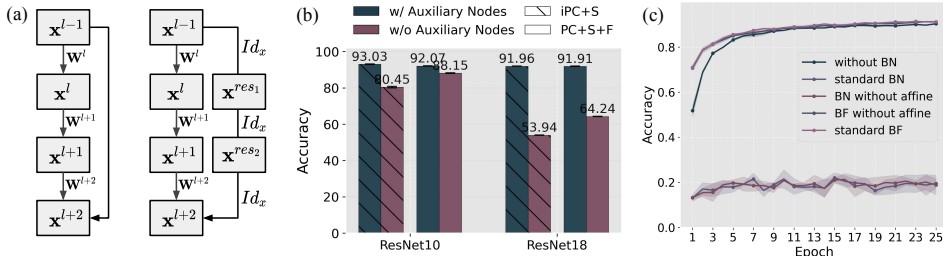

Figure 4: (a): A sketch of a residual block (left) and our proposed variation (right) with auxiliary neural activities, that prevent the error signal to travel from $\mathbf{x}^{l+2}$ to $\mathbf{x}^{l-1}$ in one timestep; (b) A barplot showing the gap in test accuracy on the CIFAR10 dataset between models with and without added neural activities on a ResNet18. Both plots refer to the best test accuracies reached by PC and iPC with the novel methods presented above. (c) Shows the test accuracy between models with the standard formulation of BN, without BN, and our proposed BF.

**BatchNorm Freezing (BF).** BatchNorm has proven instrumental in stabilizing the training of deep neural networks, as it mitigates gradient-related issues and ensures smooth gradient propagation. During training, it achieves this by normalizing layer activations through the function

$$\mathrm{BN}(x) = \gamma\left(\frac{x - \mu_B}{\sqrt{\sigma_B^2 + \epsilon}}\right) + \beta, \tag{6}$$

where $\gamma$ and $\beta$ are learnable parameters, and $\mu_B$ and $\sigma_B^2$ are the mean and variance of the minibatch $B$. At test time, it uses statistics $\mu_r$ and $\sigma_r^2$ learned through exponential moving averages. However, when applied directly to PCNs, BatchNorm fails to yield similar improvements. We hypothesize that this failure results from the iterative inference phase, where processing the same batch multiple times leads to a possible overfitting of the layer statistics. To address this issue, we propose BatchNorm Freezing (BF), a modification that freezes the states of the BatchNorm state during the inference phase, and updates running statistics exclusively during the learning phase while still using batch statistics for normalization during inference iterations.

### 4.2.1 EXPERIMENTAL VALIDATION OF THE ALGORITHMIC CONTRIBUTIONS

With the goal of validating the conjectures outlined above, we conducted two sets of experiments on the CIFAR-10 dataset: First, we trained a ResNet10 and a ResNet18 both with and without auxiliary neurons on the residual connections. In this setting, we used the two best-performing training schemes identified earlier, iPC with spiking updates, and PC with spiking plus forward updates. Second, we revisited the VGG10 architecture and compared different forms of batch normalization, including our newly proposed BF variant. The results of both experiments are reported in Fig. 4(b) and (c), respectively.

**Results.** For ResNet10, the results indicate that introducing auxiliary neurons resolves the skip-connection issue, enabling both algorithms to match the performance of the VGG10 baseline reported previously. This result is even starker in ResNet18, where the presence of multiple skip connections causes a much larger degradation of performance. Again, the addition of auxiliary neurons completely solves the problem, as the proposed models reach almost $92\%$ test accuracy, compared to a maximum of $64\%$. A similar result is obtained in the experiments with BF, where we show that normalization not only maintains convergence but also improves accuracy, indicating that stabilizing activation distributions plays a crucial role in supporting energy-based optimization within predictive coding architectures. In the next section we will show that such results, together with the ones obtained at the beginning of the section, will also extend to deeper models and more complex datasets.

## 5 LARGER SCALE EXPERIMENTS

Here, we test our proposed methods combined and show that we can reach performance comparable to that of BP when trained on models with the same complexity. To provide a comprehensive evaluation, we test them on CIFAR-10/100 (Krizhevsky et al., 2009) and TinyImageNet (Le & Yang, 2015), a

Table 2: Test accuracies of different algorithms across datasets and architectures.

| Dataset | Algorithm | VGG5 | VGG7 | VGG10 | ResNet10 | ResNet18 |
|---------|-----------|------|------|-------|----------|----------|
| CIFAR10 | BP | $90.01^{\pm 0.15}$ | $91.32^{\pm 0.14}$ | $92.68^{\pm 0.10}$ | $92.93^{\pm 0.11}$ | $93.13^{\pm 0.16}$ |
|  | PC | $87.98^{\pm 0.11}$ | $84.62^{\pm 0.10}$ | $76.22^{\pm 0.43}$ | $69.85^{\pm 2.12}$ | $15.63^{\pm 7.22}$ |
|  | iPC | $86.01^{\pm 0.10}$ | $80.15^{\pm 0.18}$ | $63.83^{\pm 0.33}$ | $62.34^{\pm 0.27}$ | $21.90^{\pm 1.51}$ |
|  | iPC + Spiking Updates | $89.73^{\pm 0.06}$ | $91.12^{\pm 0.08}$ | $93.03^{\pm 0.18}$ | $92.39^{\pm 0.04}$ | $91.96^{\pm 0.07}$ |
|  | PC + Spiking + Forward | $89.45^{\pm 0.18}$ | $90.89^{\pm 0.04}$ | $93.27^{\pm 0.10}$ | $92.47^{\pm 0.01}$ | $91.93^{\pm 0.14}$ |
| CIFAR100 (Top-1) | BP | $67.39^{\pm 0.25}$ | $67.67^{\pm 0.11}$ | $71.25^{\pm 0.21}$ | $71.21^{\pm 0.09}$ | $71.69^{\pm 0.21}$ |
|  | PC | $61.84^{\pm 0.18}$ | $56.80^{\pm 0.14}$ | $50.76^{\pm 0.37}$ | $41.51^{\pm 0.32}$ | $1.59^{\pm 0.02}$ |
|  | iPC | $56.07^{\pm 0.16}$ | $43.99^{\pm 0.30}$ | $31.99^{\pm 0.17}$ | $22.91^{\pm 0.23}$ | $1.56^{\pm 0.24}$ |
|  | iPC + Spiking Updates | $66.91^{\pm 0.12}$ | $67.10^{\pm 0.12}$ | $69.84^{\pm 0.17}$ | $70.02^{\pm 0.24}$ | $70.38^{\pm 0.20}$ |
|  | PC + Spiking + Forward | $67.16^{\pm 0.16}$ | $67.71^{\pm 0.10}$ | $72.02^{\pm 0.12}$ | $71.30^{\pm 0.21}$ | $70.90^{\pm 0.18}$ |
| CIFAR100 (Top-5) | BP | $89.56^{\pm 0.08}$ | $90.05^{\pm 0.13}$ | $92.10^{\pm 0.12}$ | $89.49^{\pm 0.13}$ | $89.43^{\pm 0.14}$ |
|  | PC | $86.53^{\pm 0.15}$ | $83.00^{\pm 0.09}$ | $78.68^{\pm 0.27}$ | $70.95^{\pm 0.35}$ | $5.89^{\pm 0.12}$ |
|  | iPC | $78.91^{\pm 0.23}$ | $73.23^{\pm 0.30}$ | $61.17^{\pm 0.31}$ | $46.41^{\pm 0.31}$ | $6.33^{\pm 0.26}$ |
|  | iPC + Spiking Updates | $89.47^{\pm 0.02}$ | $89.42^{\pm 0.06}$ | $88.75^{\pm 0.29}$ | $88.90^{\pm 0.21}$ | $90.74^{\pm 0.10}$ |
|  | PC + Spiking + Forward | $89.57^{\pm 0.09}$ | $89.62^{\pm 0.18}$ | $92.10^{\pm 0.10}$ | $89.56^{\pm 0.1}$ | $90.47^{\pm 0.11}$ |
| TinyImageNet (Top-1) | BP | $47.81^{\pm 0.12}$ | $50.13^{\pm 0.06}$ | $53.61^{\pm 0.12}$ | $53.02^{\pm 0.20}$ | $58.18^{\pm 0.12}$ |
|  | PC | $41.29^{\pm 0.20}$ | $41.15^{\pm 0.14}$ | $31.87^{\pm 0.03}$ | $13.59^{\pm 0.12}$ | $0.84^{\pm 0.02}$ |
|  | iPC | $29.94^{\pm 0.47}$ | $19.76^{\pm 0.15}$ | $11.41^{\pm 0.23}$ | $11.66^{\pm 0.39}$ | $1.44^{\pm 0.05}$ |
|  | iPC + Spiking Updates | $48.13^{\pm 0.10}$ | $48.75^{\pm 0.12}$ | $52.40^{\pm 0.20}$ | $54.94^{\pm 0.30}$ | $57.83^{\pm 0.21}$ |
|  | PC + Spiking + Forward | $49.35^{\pm 0.09}$ | $50.64^{\pm 0.12}$ | $55.31^{\pm 0.25}$ | $53.25^{\pm 0.27}$ | $54.24^{\pm 0.63}$ |
| TinyImageNet (Top-5) | BP | $72.15^{\pm 0.10}$ | $73.94^{\pm 0.10}$ | $77.11^{\pm 0.10}$ | $72.90^{\pm 0.16}$ | $79.94^{\pm 0.06}$ |
|  | PC | $66.68^{\pm 0.09}$ | $66.25^{\pm 0.11}$ | $58.14^{\pm 0.04}$ | $37.99^{\pm 0.08}$ | $5.34^{\pm 0.01}$ |
|  | iPC | $54.73^{\pm 0.52}$ | $40.36^{\pm 0.22}$ | $30.42^{\pm 0.36}$ | $26.51^{\pm 0.71}$ | $11.58^{\pm 0.21}$ |
|  | iPC + Spiking Updates | $72.71^{\pm 0.12}$ | $73.39^{\pm 0.10}$ | $76.76^{\pm 0.15}$ | $78.88^{\pm 0.32}$ | $77.55^{\pm 0.12}$ |
|  | PC + Spiking + Forward | $73.46^{\pm 0.09}$ | $75.63^{\pm 0.08}$ | $79.30^{\pm 0.17}$ | $77.72^{\pm 0.23}$ | $74.70^{\pm 0.47}$ |

scaled down version of ImageNet with 200 classes. As architectures, we use VGG-like models and ResNet architectures (He et al., 2016). Similarly to the setups of the aforementioned works, in this work we only consider models with a single feedforward layer after the convolutional layers. In all cases, we perform a large number of hyperparameter searches, and report the best test accuracy obtained with early stopping, averaged over 5 runs. We have run the experiments with and without BN/BF, and reported the best one obtained. Details on the architecture used, information needed to reproduce the results, study on hyperparameter importance, are in Appendix B.

**Results.** We report a comprehensive comparison in Table 2. As expected, in shallow models all methods can either match or approximate the performance of BP with all the methods, while in deeper models, this is not the case for our baselines, but it is for our newly proposed methods. For standard PC, spiking precisions and forward updates alone slightly improve the performance, and it is the combination of both inference and update methods that performs the best, always matching the performance of models trained with BP. To conclude, we note that batch freezing further improves the results, clearly showing that the best combination of methods is batch freezing, forward updates, and spiking precisions, which get the best results on all benchmarks when testing on a VGG10. In this case, we use models with normal BN when testing with BP.

Table 3: Test accuracies on Tiny ImageNet.

| % Accuracy | BP | PC+S+F | iPC+S |
|------------|-----|--------|-------|
| **VGG-15** |  |  |  |
| Top-1 | $44.52^{\pm 0.16}$ | $50.10^{\pm 0.16}$ | $50.24^{\pm 0.13}$ |
| Top-5 | $69.28^{\pm 0.06}$ | $73.23^{\pm 0.18}$ | $75.41^{\pm 0.10}$ |
| **VGG-15-BN/BF** |  |  |  |
| Top-1 | $53.21^{\pm 0.39}$ | $53.04^{\pm 0.36}$ | $48.19^{\pm 0.71}$ |
| Top-5 | $77.26^{\pm 0.17}$ | $76.64^{\pm 0.23}$ | $72.64^{\pm 0.20}$ |

**Scaling up.** When performing the analysis that lead to the numbers in Table 2, we noted that iPC often performed better without BF. To explicitate this phenomenon, we have performed additional experiments on a VGG15 on the Tiny ImageNet dataset. The model that we use is identical to the one proposed in the hybrid equilibrium propagation work (Nest & Ernoult, 2024). The results, reported in Table 3, show that the effect of BF is much stronger when training models with our version of PC, rather than iPC, whose performance is better when using a vanilla model, without any kind of normalization. Again, the numbers reported here are the result of an hyperparameter search, whose details are in the supplementary material.

## 6 CONCLUSION

In this work, we have investigated the following research question: Why do deep models trained with the predictive coding energy fail to match the accuracy of their counterparts trained with backpropagation? We have addressed this problem from both the algorithmic and the architectural sides. Algorithmically, we have proposed both a novel technique that leverages a dynamical precision-weighting of prediction errors to better regularize the energy landscape, and a novel weight update mechanism. From the architectural side, we have shown how to modify the residual connections to allow the training of PC-based ResNets, and developed a more effective normalization technique. The results show that we are now able to train VGG models with 15 layers, and ResNet18 on Tiny Imagenet.

A limitation of the work is the presence of the value of the predictions at initialization during the forward updates, which means that the algorithm has to store this value in memory, adding a degree of biological implausibility, due to computations not being completely local in time anymore. Future work will investigate how to address this problem with a bio-plausible weight update, with a nice starting point being a contemporaneous study that theoretically shows how to train very deep feedforward models (Innocenti et al., 2025). Despite this limitation, however, we have reached the same performance using iPC with spiking updates, and can hence claim that we have reached our goal of training complex models such as ResNet18 with a learning algorithm that is local in time and space, reaching the same performance as backprop.

## 7 REPRODUCIBILITY STATEMENT

All the experiments in this work have been run using the PCX library Pinchetti et al. (2024), an open-source software that allows training and testing predictive coding models. Besides this, all the details needed to reproduce the results are carefully described in the supplementary material. We will release the code in case of acceptance.

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

## APPENDIX

Here we provide an explanation of the performed experiments, as well as a detailed description of all the parameters needed to replicate the results of the paper. We also provide an ablation study that shows the performance of the individual methods in isolation.

**LLM Usage.** We have used LLMs for proofreading the paper and to polish writing, for retrieval and discovery of related work, and for low-level coding help (e.g. to help us produce the code for individual figures).

## A  ALGORITHM

The pseudocode is provided in Algorithm 1.

The proposed training procedure is detailed in Algorithm 1. This algorithm presents a unified framework for training both Predictive Coding (PC) and Incremental Predictive Coding (iPC) models, incorporating proposed techniques such as Precision Schedules and Forward Update. The entire process is structured into three core phases:

*Initialization:* The network's neuronal activities are initialized through a single feed-forward pass.

*Inference Learning (Relaxation Phase):* With the input and target values clamped, the hidden layer activities $(x_t^l)$ iteratively relax over T steps to minimize prediction errors across the hierarchy. For iPC, the weights $(\mathbf{W}_t^l)$ and neural activities $(x_t^l)$ are jointly updated within each iteration of this relaxation phase.

*Weight Update:* After the inference phase concludes, the PC model computes and applies its weight updates based on the final states and the initial predictions, following the Forward Update rule.

## B  EXPERIMENTS SETTING

**Model.** We conducted experiments on four VGG-based models: VGG5, VGG7, VGG10, and VGG15 and two ResNet-based models: ResNet-10, ResNet-18. The detailed architectures of these models are presented in Table 4.

**Experiments.** The benchmark results of above models are obtained with CIFAR10, CIFAR100 and Tiny ImageNet. The datasets are normalized as in Table 5.

For data augmentation on CIFAR10, CIFAR100, and Tiny ImageNet training sets, we use 50% random horizontal flipping. We also apply random cropping with different setups. For CIFAR10 and CIFAR100, images are randomly cropped to 32×32 resolution with 4-pixel padding. For Tiny ImageNet, images are randomly cropped to 64×64 resolution images with 8-pixel padding. For testing on those datasets, we applied only standard data normalization, without using any additional data augmentation techniques.

For the optimizer and scheduler, we employ mini-batch stochastic gradient descent (SGD) with momentum for updating $x$ during the relaxation phase. For the learning phase, we optimize weights $W$ using AdamW with weight decay. The learning rate schedule follows a warmup-cosine annealing pattern without restarts. This scheduler initiates training with a low learning rate during the warmup period, then smoothly transitions to a cosine-shaped decay curve, preventing abrupt performance degradation. The schedule parameters are configured as follows: the peak learning rate reaches 1.1 times the initial rate, the final learning rate settles at 0.1 times the initial rate, and the warmup phase spans 10% of the total iteration steps.

We conduct a rigorous hyperparameter search for all models, including the baselines, based on the search space specified in Table 6. All experiments were implemented using the PCX library, a JAX-based framework specifically designed for predictive coding networks that provides comprehensive benchmarking capabilities. All the experiments were conducted on NVIDIA A100/H100 GPUs, with each trial involving a hyperparameter search using the Tree-Structured Parzen Estimator (TPE) algorithm over 200 iterations. The results presented in Table 2 and Figure 3 are obtained using 5

---

**Algorithm 1** Training a PC/iPC with Precision Schedules and Forward Update

---

**Require:** Input data $\mathbf{o}$, target $\mathbf{y}$, weights $\{\mathbf{W}^l\}_{l=1}^L$, activation function $f(\cdot)$
**Require:** Weight learning rate $\eta$, activity learning rate $\alpha$, relaxation steps $T$, precision schedule $\Sigma_t^l$

---

1: **// Phase 1: Initialization**
2: $\mathbf{x}_0^0 \leftarrow \mathbf{o}$
3: **for** $l = 1$ **to** $L$ **do**          ▷ Initial feed-forward pass
4:      $\boldsymbol{\mu}_0^l \leftarrow \mathbf{W}^l f(\mathbf{x}_0^{l-1})$
5:      $\mathbf{x}_0^l \leftarrow \boldsymbol{\mu}_0^l$
6: **end for**

7: **// Phase 2: Inference Learning (Relaxation)**
8: Clamp input: $\mathbf{x}_t^0 \leftarrow \mathbf{o}$ for $t \in [1, T]$
9: Clamp output: $\mathbf{x}_t^L \leftarrow \mathbf{y}$ for $t \in [1, T]$
10: **for** $t = 0$ **to** $T - 1$ **do**
11:      **for** $l = 1$ **to** $L - 1$ **do**          ▷ Update hidden layer activities
12:          $\boldsymbol{\mu}_{t+1}^l \leftarrow \mathbf{W}^l f(\mathbf{x}_{t+1}^{l-1})$
13:          $\boldsymbol{\varepsilon}_{t+1}^l \leftarrow \mathbf{x}_t^l - \boldsymbol{\mu}_{t+1}^l$
14:          $\boldsymbol{\varepsilon}_{t+1}^{l+1} \leftarrow \mathbf{x}_t^{l+1} - \mathbf{W}^{l+1} f(\mathbf{x}_t^l)$
15:          $\Delta\mathbf{x}_t^l \leftarrow \frac{\alpha}{\Sigma_{t+1}^l} \left( \boldsymbol{\varepsilon}_{t+1}^l - (\mathbf{W}^{l+1})^\top \boldsymbol{\varepsilon}_{t+1}^{l+1} \odot f'(\mathbf{x}_t^l) \right)$
16:          $\mathbf{x}_{t+1}^l \leftarrow \mathbf{x}_t^l + \Delta\mathbf{x}_t^l$
17:          **if** iPC **then**          ▷ Update weights if in iPC Training
18:              $\Delta\mathbf{W}_{t+1}^l \leftarrow \frac{\eta}{\Sigma_{t+1}^l} \left( \boldsymbol{\varepsilon}_{t+1}^l f(\mathbf{x}_{t+1}^{l-1})^\top \right)$
19:              $\mathbf{W}_{t+1}^l \leftarrow \mathbf{W}_{t+1}^l - \Delta\mathbf{W}_{t+1}^l$
20:          **end if**
21:      **end for**
22: **end for**

23: **// Phase 3: Learning (Weight Update for PC with Forward Update)**
24: **if** PC **then**
25:      **for** $l = 1$ **to** $L$ **do**
26:          $\tilde{\boldsymbol{\varepsilon}}_T^l \leftarrow \mathbf{x}_T^l - \boldsymbol{\mu}_0^l$          ▷ Calculate Forward Update error
27:          $\Delta\mathbf{W}^l \leftarrow \eta \cdot \tilde{\boldsymbol{\varepsilon}}_T^l f(\mathbf{x}_T^{l-1})^\top$          ▷ Compute weight update
28:          $\mathbf{W}^l \leftarrow \mathbf{W}^l - \Delta\mathbf{W}^l$          ▷ Apply weight update
29:      **end for**
30: **end if**

---

different random seeds (selected from 0-4) with the optimal hyperparameter configuration. The training process is capped at 50 epochs, with an early stopping mechanism that terminates training if no accuracy improvement is observed for 10 consecutive epochs. To maintain consistency with the hyperparameter search settings, we employ a two-phase learning rate schedule: during the first 25 epochs, the weight learning rate follows a warmup-cosine-annealing schedule as previously described, after which it remains fixed at the final learning rate of the scheduler. For the results shown in Figure 2, 5 and 6, we utilize a single random seed with the optimal hyperparameters, setting the maximum training epochs to 50 without implementing early stopping. The weight learning rate schedule remains identical to the aforementioned approach.

## C   COMPUTATIONAL COMPLEXITY

In Table 7, we present the average time required to train one epoch using BP, PC, iPC, iPC with Spiking Precision (iPC + S) and PC with Spiking Precision and Forward Update (PC + S + F) across various tasks on a single H100 GPU. To eliminate the overhead associated with loading datasets into memory, we began timing from the fifth epoch onward, calculating the average duration across five consecutive epochs. We repeated this measurement process five times and report the mean and

Table 4: Detailed architectures of base models.

| VGG5 | VGG7 |
|---|---|
| Channel Sizes: [128, 256, 512, 512] | Channel Sizes: [128, 128, 256, 256, 512, 512] |
| Kernel Sizes: [3, 3, 3, 3] | Kernel Sizes: [3, 3, 3, 3, 3, 3] |
| Strides: [1, 1, 1, 1] | Strides: [1, 1, 1, 1, 1, 1] |
| Paddings: [1, 1, 1, 0] | Paddings: [1, 1, 1, 0, 1, 0] |
| Pool window: $2 \times 2$ | Pool window: $2 \times 2$ |
| Pool stride: 2 | Pool stride: 2 |
| Linear Layers: 1 | Linear Layers: 1 |

| VGG10 | VGG15 |
|---|---|
| Channel Sizes: [64, [128]x3, [256]x4, 512] | Channel Sizes: [64, 64, 128, 128, [256]x3, [512]x6] |
| Kernel Sizes: [3, 3, 3, 3, 3, 3, 3, 3, 3] | Kernel Sizes: [3, 3, 3, 3, 3, 3, 3, 3, 3, 3, 3, 3, 3] |
| Strides: [1, 1, 1, 1, 1, 1, 1, 1, 1] | Strides: [1, 1, 1, 1, 1, 1, 1, 1, 1, 1, 1, 1, 1] |
| Paddings: [1, 1, 1, 1, 1, 1, 1, 1, 1] | Paddings: [1, 1, 1, 1, 1, 1, 1, 1, 1, 1, 1, 1, 1] |
| Pool window: $2 \times 2$ | Pool window: $2 \times 2$ |
| Pool stride: 2 | Pool stride: 2 |
| Linear Layers: 1 | Linear Layers: 2 |

| ResNet10 | ResNet18 |
|---|---|
| Initial Conv: 3x3, Stride 1, Channel Size 64 | Initial Conv: 3x3, Stride 1, Channel Size 64 |
| Res-Block: [1, 1, 1, 1] | Res-Block: [2, 2, 2, 2] |
| Channel Sizes: [64, 128, 256, 512] | Channel Sizes: [64, 128, 256, 512] |
| Strides: [1, 2, 2, 2] | Strides: [1, 2, 2, 2] |
| Linear Layers: 1 | Linear Layers: 1 |

Table 5: Data normalization.

| | Mean ($\mu$) | Std ($\sigma$) |
|---|---|---|
| CIFAR10 | [0.4914, 0.4822, 0.4465] | [0.2023, 0.1994, 0.2010] |
| CIFAR100 | [0.5071, 0.4867, 0.4408] | [0.2675, 0.2565, 0.2761] |
| Tiny ImageNet | [0.485, 0.456, 0.406] | [0.229, 0.224, 0.225] |

standard deviation of these five experimental runs. It is worth noting that the reported times for predictive coding suffer from an implementation bottleneck: despite the possibility of updating all the neural activities in parallel, our library does not allow that. This largely slows down our models when trained on deep architectures.

**Results.** The results in Table 7 lead to several key observations. First, our proposed methods, PC+S+F and iPC+S, exhibit training times nearly identical to their respective baselines, PC and iPC. This demonstrates that the proposed spiking precision and forward update mechanisms do not introduce a significant computational overhead.

The table also shows that iPC is consistently slower than PC. This performance difference stems from their distinct weight update strategies. PC first runs the inference learning for T timesteps to allow the neural activities ($x$) to converge and then performs a single weight update at the end. In contrast, iPC performs weight updates within each of the T inference steps, alongside the neural activity updates. This approach results in T separate weight updates for every layers instead of one, is the direct cause of its higher computational cost compared to PC.

Finally, all PC models are slower than BP, with this ratio increasing as the number of model layers increases, mostly for the bottleneck just described. While the forward pass is computationally identical for both BP and PC, their backward/update passes differ fundamentally. BP computes gradients in a single backward pass. In contrast, PC perform an iterative inference process to update neural activities by minimizing a global prediction error. This process runs for a fixed number of T. The computational complexity of a single BP epoch is proportional to the number of layers, L. For PC, each of the T inference steps involves computations across all layers, making their complexity roughly proportional to T×L. Since optimal performance often requires T to be equal to or greater than the network depth L, the computational cost for PC naturally scales more rapidly with deeper

Table 6: Hyperparameters search configuration.

| Parameter | PCNs | BP |
|---|:---:|:---:|
| Epoch | 25 | |
| Batch Size | 128 | |
| Activation | [leaky relu, gelu, hard tanh, relu] | |
| $\alpha$ | [0.01, 0.05, 0.001, 0.005] | - |
| $\beta$ | [0.0, 1.0], 0.15[1] | - |
| $lr_x*$ | (5e-3, 9e-1)[2] | - |
| $lr_w$ | (1e-5, 3e-2)[2] | (1e-5, 3e-4)[2] |
| $momentum_x$ | [0.0, 1.0], 0.1[1] | - |
| $weight\_decay_w$ | (1e-5, 1e-2)[2] | |
| T (VGG-5) | [5,6,7,8] | - |
| T (VGG-7) | [7,9,11,13] | - |
| T (VGG-10) | [10,12,14,16] | - |
| T (VGG-15) | [15,17,19,21] | - |
| T (ResNet-10) | [10,12,14,16] | - |
| T (ResNet-18) | [18,20,22,24] | - |

[1]: "[a, b], c" denotes a sequence of values from a to b with a step size of c. [2]: "(a, b)" represents a log-uniform distribution between a and b.

Table 7: Comparison of the training times (seconds per epoch) of BP against PCNs on different architectures with CIFAR10

| Task | BP | PC | PC + S + F | iPC | iPC + S |
|---|:---:|:---:|:---:|:---:|:---:|
| VGG5 (T = 5) | $1.16^{\pm 0.02}$ | $1.56^{\pm 0.01}$ | $1.55^{\pm 0.01}$ | $1.94^{\pm 0.01}$ | $1.94^{\pm 0.01}$ |
| VGG7 (T = 7) | $1.29^{\pm 0.02}$ | $2.20^{\pm 0.01}$ | $2.20^{\pm 0.01}$ | $2.94^{\pm 0.01}$ | $2.95^{\pm 0.01}$ |
| VGG10 (T = 10) | $1.90^{\pm 0.01}$ | $5.08^{\pm 0.02}$ | $5.06^{\pm 0.02}$ | $7.00^{\pm 0.05}$ | $7.01^{\pm 0.03}$ |
| ResNet10 (T = 10) | $1.75^{\pm 0.01}$ | $5.02^{\pm 0.02}$ | $5.10^{\pm 0.01}$ | $7.08^{\pm 0.05}$ | $7.08^{\pm 0.05}$ |
| ResNet18 (T = 18) | $2.78^{\pm 0.01}$ | $14.92^{\pm 0.01}$ | $15.17^{\pm 0.03}$ | $22.14^{\pm 0.05}$ | $22.14^{\pm 0.01}$ |

architectures, despite this not being as much of a bottleneck as the full parallelization of the operations. Thus, We expect predictive coding networks to maintain computational efficiency across larger model architectures while offering substantial performance advantages when implemented on specialized analog neuromorphic hardware.

# D  ABLATION STUDY

In this section, we conduct ablation studies to evaluate the individual and synergistic effects of our proposed components. By systematically isolating each mechanism, we quantify its specific contribution to the overall performance of the model.

## D.1  SPIKING PRECISION

### D.1.1  COMPARISON OF DIFFERENT PRECISION SCHEDULES

Our central hypothesis is that mitigating the energy imbalance in deep networks requires a potent and precisely timed signal amplification. This amplification should occur at the moment the error information, propagating from the output layer, first arrives at a given hidden layer. To test this hypothesis, we designed and compared several dynamic precision schedules.

Based on this perspective, in addition to Spiking Precision, we designed a Decaying Precision schedule, which offers a slightly smoother, yet still powerful, amplification profile. The formula for Decaying Precision is as follows:

$$\Sigma_t^l = \begin{cases} \frac{\sum_{j=0}^{T-L+l} e^{-k \cdot j}}{e^{-k \cdot (l-L+t)}}, & \text{when } l \geq L - t, \\ 1, & \text{when } l < L - t. \end{cases} \tag{7}$$

Table 8: Test accuracies of the different algorithms across architectures and datasets.

| Dataset | Algorithm | VGG5 | VGG7 | VGG10 | VGG5BF | VGG7BF | VGG10BF |
|---|---|---|---|---|---|---|---|
| CIFAR10 | BP | $89.43^{\pm 0.12}$ | $89.91^{\pm 0.12}$ | $\mathbf{92.21^{\pm 0.08}}$ | $90.01^{\pm 0.15}$ | $91.32^{\pm 0.14}$ | $92.68^{\pm 0.10}$ |
| | PC | $87.98^{\pm 0.11}$ | $84.62^{\pm 0.10}$ | $72.75^{\pm 6.03}$ | $87.77^{\pm 0.14}$ | $80.62^{\pm 0.14}$ | $76.22^{\pm 0.43}$ |
| | Decaying Precision (D) | $87.91^{\pm 0.22}$ | $81.14^{\pm 0.19}$ | $84.87^{\pm 0.19}$ | $88.55^{\pm 0.09}$ | $80.00^{\pm 0.10}$ | $81.68^{\pm 0.18}$ |
| | Spiking Precision (S) | $88.06^{\pm 0.16}$ | $82.16^{\pm 0.14}$ | $78.09^{\pm 0.61}$ | $88.53^{\pm 0.07}$ | $86.51^{\pm 0.21}$ | $83.17^{\pm 0.07}$ |
| | PC+D+F | $89.32^{\pm 0.14}$ | $89.34^{\pm 0.09}$ | $89.43^{\pm 0.18}$ | $\mathbf{90.37^{\pm 0.13}}$ | $\mathbf{91.48^{\pm 0.12}}$ | $91.46^{\pm 0.12}$ |
| | PC+S+F | $\mathbf{89.45^{\pm 0.18}}$ | $\mathbf{90.08^{\pm 0.21}}$ | $92.07^{\pm 0.10}$ | $89.30^{\pm 0.13}$ | $90.89^{\pm 0.04}$ | $\mathbf{93.27^{\pm 0.10}}$ |
| CIFAR100 (Top-1) | BP | $66.28^{\pm 0.23}$ | $65.36^{\pm 0.15}$ | $\mathbf{69.35^{\pm 0.16}}$ | $67.39^{\pm 0.25}$ | $67.67^{\pm 0.11}$ | $71.25^{\pm 0.21}$ |
| | PC | $60.00^{\pm 0.19}$ | $56.80^{\pm 0.14}$ | $45.86^{\pm 1.70}$ | $61.84^{\pm 0.18}$ | $55.57^{\pm 0.14}$ | $50.76^{\pm 0.37}$ |
| | Decaying Precision (D) | $57.76^{\pm 0.33}$ | $45.05^{\pm 0.37}$ | $55.66^{\pm 0.88}$ | $66.05^{\pm 0.12}$ | $51.11^{\pm 0.32}$ | $53.27^{\pm 0.48}$ |
| | Spiking Precision (S) | $59.18^{\pm 0.20}$ | $56.98^{\pm 0.19}$ | $51.56^{\pm 0.16}$ | $60.34^{\pm 0.28}$ | $55.74^{\pm 0.15}$ | $56.24^{\pm 0.37}$ |
| | PC+D+F | $66.10^{\pm 0.09}$ | $64.86^{\pm 0.10}$ | $66.54^{\pm 0.12}$ | $\mathbf{67.56^{\pm 0.25}}$ | $67.27^{\pm 0.21}$ | $69.81^{\pm 0.22}$ |
| | PC+S+F | $\mathbf{66.49^{\pm 0.15}}$ | $\mathbf{66.34^{\pm 0.22}}$ | $69.08^{\pm 0.08}$ | $67.16^{\pm 0.16}$ | $\mathbf{67.71^{\pm 0.10}}$ | $\mathbf{72.02^{\pm 0.12}}$ |
| CIFAR100 (Top-5) | BP | $85.85^{\pm 0.27}$ | $84.41^{\pm 0.26}$ | $\mathbf{88.74^{\pm 0.08}}$ | $89.56^{\pm 0.08}$ | $\mathbf{90.05^{\pm 0.13}}$ | $92.10^{\pm 0.12}$ |
| | PC | $84.97^{\pm 0.19}$ | $83.00^{\pm 0.09}$ | $74.61^{\pm 1.08}$ | $86.53^{\pm 0.15}$ | $82.07^{\pm 0.35}$ | $78.68^{\pm 0.27}$ |
| | Decaying Precision (D) | $81.59^{\pm 0.13}$ | $74.00^{\pm 0.30}$ | $83.13^{\pm 0.74}$ | $88.82^{\pm 0.07}$ | $78.93^{\pm 0.29}$ | $81.16^{\pm 0.36}$ |
| | Spiking Precision (S) | $84.58^{\pm 0.12}$ | $83.61^{\pm 0.15}$ | $78.62^{\pm 0.15}$ | $85.86^{\pm 0.10}$ | $82.64^{\pm 0.14}$ | $83.44^{\pm 0.21}$ |
| | PC+D+F | $85.85^{\pm 0.10}$ | $83.80^{\pm 0.20}$ | $86.10^{\pm 0.21}$ | $\mathbf{89.84^{\pm 0.17}}$ | $89.74^{\pm 0.12}$ | $91.24^{\pm 0.07}$ |
| | PC+S+F | $\mathbf{86.36^{\pm 0.11}}$ | $\mathbf{84.53^{\pm 0.15}}$ | $86.84^{\pm 0.07}$ | $89.57^{\pm 0.09}$ | $89.62^{\pm 0.18}$ | $92.10^{\pm 0.10}$ |

Table 9: Test accuracies of the different algorithms on Tiny ImageNet.

| Algorithm | Top-1 Accuracy | | Top-5 Accuracy | |
|---|---|---|---|---|
| | VGG15 | VGG15BF | VGG15 | VGG15BF |
| PC+S+Forward Update | $42.51^{\pm 0.18}$ | $53.04^{\pm 0.36}$ | $66.22^{\pm 0.18}$ | $76.64^{\pm 0.23}$ |
| PC | $22.95^{\pm 1.50}$ | $22.91^{\pm 0.61}$ | $47.04^{\pm 2.04}$ | $45.64^{\pm 0.63}$ |
| Decaying Precision (D) | $27.29^{\pm 0.24}$ | $22.05^{\pm 0.16}$ | $54.18^{\pm 0.37}$ | $45.22^{\pm 0.23}$ |
| Spiking Precision (S) | $18.36^{\pm 0.36}$ | $17.95^{\pm 0.14}$ | $39.24^{\pm 0.31}$ | $39.87^{\pm 0.51}$ |
| PC+D+Forward Update | $21.95^{\pm 0.20}$ | $30.83^{\pm 0.77}$ | $45.15^{\pm 0.23}$ | $56.06^{\pm 0.85}$ |

Here, the numerator sum serves as a normalization term that ensures that the sum of the layer-wise precisions over time is equal to one, that is, $\sum_{t=1}^{T}(\Sigma_t^l)^{-1} = 1$. The denominator $e^{-k \cdot (l-L+t)}$ allows lower layers to receive larger weights when activated ($l \geq L - t$), thereby helping to achieve a more balanced energy distribution during the inference phase, k is a hyperparameter that controls the strength of this balancing effect, the search range is $[1.0, 1.5, 2.0]$. It also ensures that each layer experiences a significant boost in precision precisely when the energy from the output first reaches that layer ($l = L - t$). When $l < L - t$, we set $\Sigma_t^l = 1$.

As shown in Table 8, both Decaying and Spiking Precision schedules offer improvements over the baseline PC, particularly in deeper models like VGG10. However, a clear pattern emerges when we analyze their effectiveness in relation to network depth. In shallower models like VGG5 and VGG7, the performance of Decaying Precision is comparable to that of Spiking Precision. This suggests that when the signal path is short, a moderately amplification is sufficient.

As shown in the Tab. 9, when model depth increases to VGG15 with TinyImageNet task, a noticeable performance gap appears, with Spiking Precision consistently outperforming Decaying Precision. This finding strongly supports our core hypothesis: the exponential signal attenuation in deeper networks necessitates a correspondingly sharp and powerful counteracting signal. The abrupt, targeted amplification of Spiking Precision is more effective at preserving the integrity of the error signal across many layers than the smoother profile of Decaying Precision. Consequently, for all subsequent experiments reported in the main body of this paper, we exclusively utilized the superior Spiking Precision schedule. This investigation also opens exciting avenues for future work, such as exploring hybrid schedules that might combine the strengths of different amplification profiles.

### D.1.2 ENERGY PROPAGATION WITH PRECISION

We observed that removing the decaying/spiking precision module consistently leads to performance degradation. This effect is particularly evident in deeper models like VGG7 and VGG10, where its absence causes a significant imbalance in the energy distribution across layers. For instance,

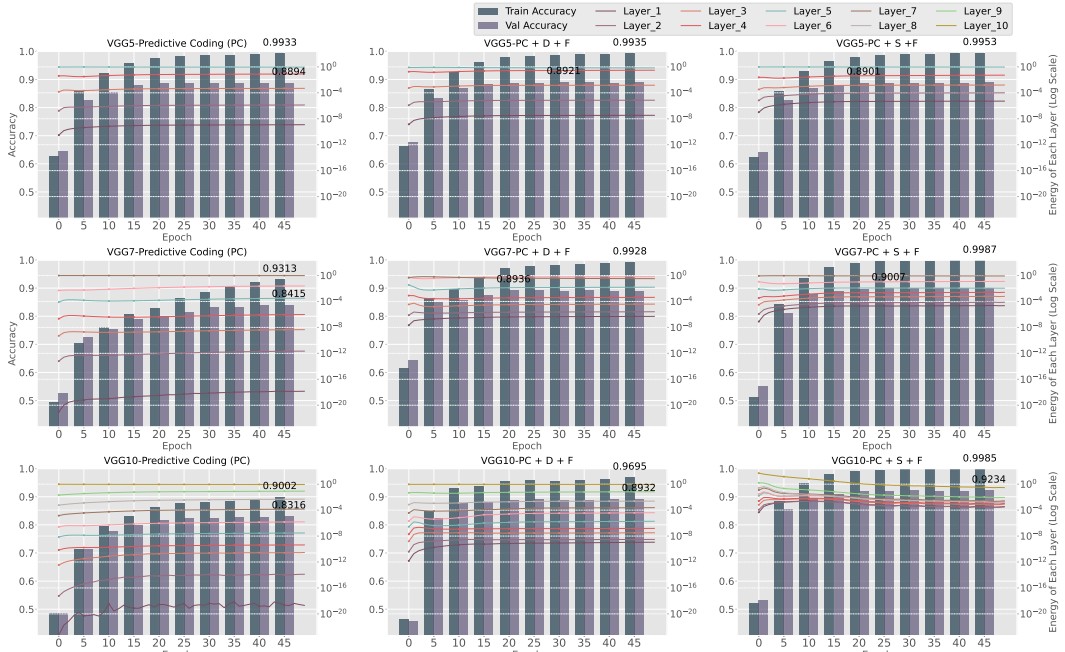

Figure 5: Layer-wise Energy Distribution and Accuracy Comparison between PC and Decaying Precision/Spiking Precision with Forward Update in VGG5, VGG7 and VGG10 on the CIFAR10 dataset. The colored lines represent the total energy of the individual layers of the model. The vertical lines represent the train and test accuracies of the model.

as shown in Figure 5, the energy proportion of the first layer in the VGG7 model with spiking precision is approximately $10^{-6}$. Without this precision term, the proportion plummets to $10^{-18}$. A comparison of the layer-wise energy distributions (Figure 6) confirms that our proposed precision methods effectively rebalance energy propagation. By increasing the energy in the initial layers by several orders of magnitude, these methods rectify the imbalance, which contributes directly to improved model performance.

Furthermore, the degree of this energy rebalancing correlates with the performance difference between the Decaying and Spiking Precision variants. In the VGG5 and VGG7 models, where the accuracy gap between the two methods is minimal, the difference in their first-layer energy distributions is also small. However, in VGG10, where Spiking Precision significantly outperforms Decaying Precision, the energy gap is far more pronounced. Specifically, the first layer's energy proportion is approximately $10^{-10}$ for Decaying Precision, whereas Spiking Precision elevates it to $10^{-4}$, highlighting a clear link between balanced energy propagation and model accuracy.

## D.2 FORWARD UPDATE (FU)

### D.2.1 NEURAL ACTIVITY DIVERGENCY QUANTIFICATION

To quantify the neural activity divergence that Forward Update aims to solve, we conducted a new experiment on a VGG10 model trained on CIFAR-10. We measured the Mean Squared Error between the initial and final neural states, $MSE(x_0^l, x_T^l)$, at each weight update. We then calculated the ratio of the square root of this divergence to the energy used for the weight update in that layer. We term this metric the "Gap Ratio". As shown in Table 10, in the model trained without Forward Update, the Gap Ratio in the final layer (L10) is extremely large and unstable across epochs, indicating that the neural activity divergence completely dominates the weight update signal. This supports our hypothesis that this divergence causes errors to accumulate in the final layers, destabilizing learning. In contrast, the model trained with Forward Update (Table 11) shows a dramatically reduced and stable Gap Ratio in the final layer.

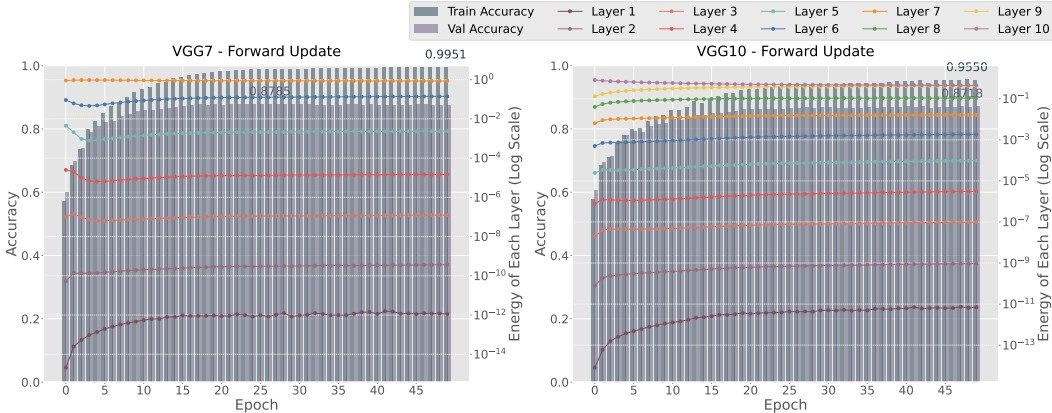

Figure 6: Layer-wise Energy Distribution and Accuracy of Forward Update in VGG5, VGG7 and VGG10 on the CIFAR10 dataset. The colored lines represent the total energy of the individual layers of the model. The vertical lines represent the train and test accuracies of the model.

Table 10: Gap Ratio on VGG10/CIFAR10 trained without Forward Update.

| w/o FU | L1 | L2 | L3 | L4 | L5 | L6 | L7 | L8 | L9 | L10 | Train Acc. | Test Acc. |
|---|---|---|---|---|---|---|---|---|---|---|---|---|
| Epoch 5 | 0.0 | 99.49 | 98.98 | 97.08 | 106.78 | 113.52 | 110.77 | 468.9 | 566.52 | 7065.72 | 73.75 | 74.83 |
| Epoch 15 | 0.0 | 99.51 | 99.4 | 97.2 | 98.95 | 105.62 | 132.44 | 108.39 | 101.0 | 3807.9 | 51.86 | 51.9 |
| Epoch 25 | 0.0 | 99.25 | 97.94 | 98.22 | 89.95 | 90.48 | 116.55 | 113.2 | 102.21 | 4716.88 | 61.21 | 62.3 |
| Epoch 35 | 0.0 | 99.41 | 98.24 | 96.97 | 79.12 | 95.99 | 122.38 | 106.3 | 104.07 | 6498.41 | 60.31 | 60.14 |
| Epoch 45 | 0.0 | 98.23 | 97.94 | 97.36 | 80.84 | 90.42 | 116.96 | 108.26 | 106.42 | 5362.02 | 59.87 | 61.42 |

Table 11: Gap Ratio on VGG10/CIFAR10 trained with Forward Update.

| with FU | L1 | L2 | L3 | L4 | L5 | L6 | L7 | L8 | L9 | L10 | Train Acc. | Test Acc. |
|---|---|---|---|---|---|---|---|---|---|---|---|---|
| Epoch 5 | 0.0 | 292.58 | 112.77 | 224.2 | 120.89 | 346.42 | 143.95 | 248.32 | 100.78 | 5.14 | 88.27 | 85.66 |
| Epoch 15 | 0.0 | 327.43 | 132.66 | 265.24 | 151.09 | 409.26 | 264.34 | 545.12 | 117.6 | 8.27 | 97.97 | 90.99 |
| Epoch 25 | 0.0 | 323.0 | 135.82 | 256.2 | 149.77 | 407.2 | 269.14 | 619.95 | 115.07 | 9.43 | 99.55 | 92.08 |
| Epoch 35 | 0.0 | 322.42 | 138.05 | 258.38 | 148.22 | 415.02 | 292.98 | 681.21 | 115.12 | 10.06 | 99.73 | 92.04 |
| Epoch 45 | 0.0 | 323.25 | 135.72 | 261.67 | 149.45 | 417.04 | 312.4 | 731.19 | 115.87 | 10.54 | 99.83 | 92.22 |

To further isolate the effect of Forward Update, we performed an additional ablation where FU was applied only to the final three layers during the weight update, without any additional hyperparameter tuning. The results in Table 12 show that even this targeted application of Forward Update yields significant improvements in both training stability and final accuracy compared to the baseline. This reinforces that addressing the neural activity divergence in the deepest layers is a critical factor for success.

Table 12: Gap Ratio on VGG10/CIFAR10 when applying Forward Update only to the last 3 layers (L8, L9, L10).

| | L1 | L2 | L3 | L4 | L5 | L6 | L7 | L8 | L9 | L10 | Train Acc. | Test Acc. |
|---|---|---|---|---|---|---|---|---|---|---|---|---|
| Epoch 5 | 0.0 | 96.21 | 57.22 | 50.78 | 32.03 | 53.59 | 30.16 | 49.77 | 66.91 | 42.19 | 72.75 | 74.13 |
| Epoch 15 | 0.0 | 95.51 | 57.52 | 49.76 | 31.65 | 56.92 | 32.96 | 40.87 | 70.08 | 62.02 | 79.23 | 78.56 |
| Epoch 25 | 0.0 | 95.58 | 59.17 | 46.63 | 16.04 | 91.18 | 34.19 | 39.63 | 73.17 | 67.89 | 81.4 | 80.14 |
| Epoch 35 | 0.0 | 95.44 | 56.84 | 29.72 | 9.65 | 94.43 | 35.13 | 39.37 | 72.12 | 70.17 | 80.91 | 79.97 |
| Epoch 45 | 0.0 | 95.51 | 56.88 | 37.85 | 15.11 | 93.49 | 34.8 | 38.59 | 71.82 | 72.16 | 80.02 | 79.4 |

### D.2.2 MODEL ROBUSTNESS ANALYSIS

Predictive Coding (PC) networks are often noted for their inherent robustness compared to networks trained with Backpropagation Salvatori et al. (2021). Unlike our precision-weighting mechanisms, the Forward Update (F) method alters the core computational diagram of PC. Therefore, we conducted experiments to investigate whether this modification adversely affects model robustness.

Our first experiment provides a fair comparison on a VGG5 model, where standard PC and our PC+F variant achieve similar baseline accuracies on clean data. We trained both models on CIFAR-10 and evaluated their calibration under six types of data corruption across five levels of intensity. The range of the Adaptive Expected Calibration Error (adaECE) is reported in Table 13. Note that to ensure a meaningful ECE calculation, we scale the output logits before applying the softmax function. The results indicate that our PC+F model maintains a robustness profile that is comparable to, and at higher corruption levels, superior to that of standard PC.

Table 13: AdaECE range at different levels of corruption using VGG5.

| Corruption Level | PC+FU | PC |
|---|---|---|
| 0.1 | [0.091665, 0.327372] | [0.037817, 0.388640] |
| 0.2 | [0.089805, 0.349025] | [0.036528, 0.406428] |
| 0.3 | [0.034430, 0.363157] | [0.035065, 0.419023] |
| 0.4 | [0.021906, 0.375255] | [0.029735, 0.428372] |
| 0.5 | [0.026589, 0.383087] | [0.033315, 0.434768] |

Furthermore, to evaluate our methods on deeper models, we compared ourPC+S+FU model against the BP baseline on the VGG10 architecture. As shown in Table 14, our method exhibits a robustness profile that is highly comparable to backpropagation across all tested corruption intensities. Taken together, these experiments demonstrate that our proposed methods not only enable the training of deeper PCNs but do so while preserving the desirable property of model robustness.

Table 14: AdaECE range at different levels of corruption using VGG10.

| Corruption Level | BP | PC+S+FU |
|---|---|---|
| 0.1 | [0.039071, 0.485300] | [0.039269, 0.486342] |
| 0.2 | [0.043376, 0.490541] | [0.045973, 0.486978] |
| 0.3 | [0.043943, 0.494404] | [0.047871, 0.486432] |
| 0.4 | [0.067549, 0.496167] | [0.077340, 0.486966] |
| 0.5 | [0.132682, 0.499181] | [0.143458, 0.486497] |

### D.2.3 ON THE BIOLOGICAL PLAUSIBILITY OF FORWARD UPDATE

A core motivation for using PCNs over backpropagation is their biological plausibility — particularly their use of local learning rules and temporally local computations. The Forward Update (F) mechanism, while effective, seems to introduce non-locality in time by requiring each synapse to store its initial feedforward activity $\mu_0^l$ until convergence.

However, the description of FU in our manuscript was chosen for conceptual clarity; it is not a fundamental requirement of the method. In practice, the initial feed-forward state can be re-computed through a temporally local and biologically plausible process. This is achieved by introducing a "free relaxation" phase after the inference learning and before the weight update. In this phase, the label clamp is removed, and the network settles to a new equilibrium with only the sensory input clamped, just as in Equilibrium Propagation (Ernoult et al., 2020) before the nudging phase.

In this setting, the network naturally converges to the state corresponding to its feed-forward prediction (Frieder & Lukasiewicz, 2022). Since the weights have not yet been updated, this re-computed state is identical to the $\mu_0^l$ in our formulation. This eliminates the need for long-term storage and resolves the concern of temporal non-locality. The primary contribution of our F method is that it identifies and solves a critical failure mode in deep predictive coding networks: the accumulation of errors caused by the divergence of neural activities from their initial predictions. While the biologically plausible implementation we describe requires extra computation, future work can focus on developing mechanisms that are both fully plausible and computationally efficient for PC.

Table 15: Test accuracies of different algorithms without BatchNorm Freeze across datasets and architectures.

| Dataset | Algorithm | VGG5 | VGG7 | VGG10 | ResNet10 | ResNet18 |
|---|---|---|---|---|---|---|
| CIFAR10 | BP | $89.43^{\pm 0.12}$ | $89.91^{\pm 0.12}$ | $92.21^{\pm 0.08}$ | $92.21^{\pm 0.20}$ | $\mathbf{92.32^{\pm 0.22}}$ |
| | PC | $87.98^{\pm 0.11}$ | $84.62^{\pm 0.10}$ | $72.75^{\pm 6.03}$ | $69.85^{\pm 2.12}$ | $15.63^{\pm 7.22}$ |
| | iPC | $85.51^{\pm 0.12}$ | $80.15^{\pm 0.18}$ | $63.83^{\pm 0.33}$ | $62.34^{\pm 0.27}$ | $21.90^{\pm 1.51}$ |
| | PC+Spiking Precision (S) | $88.06^{\pm 0.16}$ | $82.16^{\pm 0.14}$ | $78.09^{\pm 0.61}$ | $80.61^{\pm 0.20}$ | $80.07^{\pm 0.21}$ |
| | PC+Forward Update (F) | $88.79^{\pm 0.04}$ | $87.43^{\pm 0.30}$ | $87.33^{\pm 0.14}$ | $66.94^{\pm 0.76}$ | $25.50^{\pm 3.12}$ |
| | iPC+S | $89.32^{\pm 0.13}$ | $\mathbf{90.25^{\pm 0.06}}$ | $\mathbf{93.03^{\pm 0.18}}$ | $\mathbf{92.39^{\pm 0.04}}$ | $91.96^{\pm 0.07}$ |
| | PC+S+F | $\mathbf{89.45^{\pm 0.18}}$ | $90.08^{\pm 0.21}$ | $92.07^{\pm 0.10}$ | $92.04^{\pm 0.04}$ | $91.91^{\pm 0.09}$ |
| CIFAR100 (Top-1) | BP | $66.28^{\pm 0.23}$ | $65.36^{\pm 0.15}$ | $69.35^{\pm 0.16}$ | $69.23^{\pm 0.09}$ | $\mathbf{71.46^{\pm 0.12}}$ |
| | PC | $60.00^{\pm 0.19}$ | $56.80^{\pm 0.14}$ | $45.86^{\pm 1.70}$ | $27.62^{\pm 3.03}$ | $1.59^{\pm 0.02}$ |
| | iPC | $56.07^{\pm 0.16}$ | $43.99^{\pm 0.30}$ | $21.37^{\pm 0.37}$ | $22.91^{\pm 0.23}$ | $1.53^{\pm 0.06}$ |
| | PC+Spiking Precision (S) | $59.18^{\pm 0.20}$ | $56.98^{\pm 0.19}$ | $51.56^{\pm 0.16}$ | $50.23^{\pm 0.20}$ | $22.92^{\pm 0.15}$ |
| | PC+Forward Update (F) | $65.34^{\pm 0.07}$ | $64.50^{\pm 0.14}$ | $61.69^{\pm 0.79}$ | $39.89^{\pm 0.90}$ | $3.42^{\pm 0.10}$ |
| | iPC+S | $65.54^{\pm 0.62}$ | $65.76^{\pm 0.12}$ | $\mathbf{69.84^{\pm 0.17}}$ | $\mathbf{70.02^{\pm 0.24}}$ | $70.38^{\pm 0.20}$ |
| | PC+S+F | $\mathbf{66.49^{\pm 0.15}}$ | $\mathbf{66.34^{\pm 0.22}}$ | $69.08^{\pm 0.08}$ | $68.99^{\pm 0.18}$ | $70.81^{\pm 0.08}$ |
| CIFAR100 (Top-5) | BP | $85.85^{\pm 0.27}$ | $84.41^{\pm 0.26}$ | $\mathbf{88.74^{\pm 0.08}}$ | $87.75^{\pm 0.10}$ | $89.43^{\pm 0.14}$ |
| | PC | $84.97^{\pm 0.19}$ | $83.00^{\pm 0.09}$ | $74.61^{\pm 1.08}$ | $57.93^{\pm 2.62}$ | $5.89^{\pm 0.12}$ |
| | iPC | $78.91^{\pm 0.23}$ | $73.23^{\pm 0.30}$ | $48.35^{\pm 0.79}$ | $46.41^{\pm 0.31}$ | $6.33^{\pm 0.26}$ |
| | PC+Spiking Precision (S) | $84.58^{\pm 0.12}$ | $83.61^{\pm 0.15}$ | $78.62^{\pm 0.15}$ | $77.84^{\pm 0.23}$ | $53.89^{\pm 0.06}$ |
| | PC+Forward Update (F) | $85.48^{\pm 0.08}$ | $84.05^{\pm 0.07}$ | $76.73^{\pm 0.92}$ | $69.48^{\pm 0.70}$ | $15.25^{\pm 0.04}$ |
| | iPC+S | $85.66^{\pm 0.29}$ | $\mathbf{84.96^{\pm 0.14}}$ | $88.70^{\pm 0.18}$ | $\mathbf{88.90^{\pm 0.21}}$ | $90.05^{\pm 0.20}$ |
| | PC+S+F | $\mathbf{86.36^{\pm 0.11}}$ | $84.53^{\pm 0.15}$ | $88.66^{\pm 0.14}$ | $88.45^{\pm 0.16}$ | $\mathbf{90.47^{\pm 0.11}}$ |

### D.3 ABLATION STUDY ON FORWARD UPDATE AND SPIKING PRECISION

To dissect the individual and combined contributions of our primary algorithmic modifications, Spiking Precision (S) and Forward Update (F), we conducted a detailed ablation study, the results of which are presented in Table 15. This analysis systematically evaluates each component's impact on both standard PC and iPC across various architectures and datasets, excluding the effects of BatchNorm Freezing to isolate the core mechanisms.

The results reveal a clear and complementary relationship between Spiking Precision and Forward Update for standard PC. When applied in isolation, Forward Update (PC+F) significantly improves performance on VGG-style architectures, stabilizing training and preventing the sharp accuracy degradation seen in the baseline PC as depth increases. For instance, on CIFAR10, PC+F maintains an accuracy of around $87\%$ on VGG10, whereas the baseline PC drops to $72.75\%$. However, Forward Update alone is insufficient for training deep residual networks; its performance on ResNet18 is only marginally better than the baseline, failing to overcome the catastrophic failure. This suggests that while F effectively mitigates weight update divergence, it does not solve the underlying problem of energy imbalance in architectures with skip connections.

Conversely, Spiking Precision alone (PC+S) offers a substantial improvement on ResNet models, preventing the complete collapse of training. On ResNet18 with CIFAR10, it achieves an accuracy of $80.07\%$, a dramatic recovery from the baseline's $15.63\%$. This confirms its crucial role in rebalancing energy and ensuring a viable error signal reaches the early layers in models with skip-connections. However, on its own, it does not elevate performance to the level of backpropagation.

The true strength of our approach is demonstrated when both components are combined. The PC+S+F model consistently achieves performance on par with, and occasionally exceeding, backpropagation across all tested architectures, including the challenging ResNet18. This powerful synergy underscores that both mechanisms are essential: Spiking Precision addresses the signal propagation problem, while Forward Update addresses the update stability problem.

Interestingly, for iPC, the addition of Spiking Precision alone (iPC+S) is sufficient to achieve state-of-the-art performance, rivaling both BP and the fully-equipped PC+S+F. This indicates that the incremental nature of iPC, where weights are updated at every inference timestep, inherently prevents the large divergence between forward neural state and backward neural states that Forward Update is designed to correct. With its continuous adaptation, iPC only requires the energy rebalancing provided by Spiking Precision to successfully train deep architectures.

## D.4 BATCHNORM FREEZING (BF)

To isolate the precise source of instability when applying BatchNorm (BN) to Predictive Coding Networks (PCNs), we sought to determine whether the problem stems from the learnable affine parameters $(\gamma, \beta)$ or from the iterative updating of batch statistics $(\mu_B, \sigma_B^2)$ during the inference phase. While the affine parameters can be a source of overfitting in standard training, we hypothesized that the unique, multi-step inference process of PCNs creates a different challenge: the repeated processing of a single mini-batch causes the batch statistics themselves to overfit, destabilizing the network dynamics. Our BatchNorm Freezing (BF) method is designed specifically to solve this issue.

To test this hypothesis, we conducted an ablation study on the VGG10-CIFAR10 task. The results presented in Table 16, provide clear evidence for our claim. Removing the affine parameters from standard BN still had a negligible effect on the performance degradation (24.31% and 24.28%), indicating that these parameters are not the source of the problem. In contrast, freezing the statistics during inference improved performance over the without batch normalization baseline. This confirms that the iterative updates to batch statistics are the primary cause of instability and that our proposed BF method is an effective solution.

Table 16: Ablation study on different BatchNorm strategies (VGG10-CIFAR10).

| Method | Test Accuracy (%) |
|---|---|
| Without BN | $92.07 \pm 0.10$ |
| Standard BN | $24.31 \pm 4.51$ |
| BN without affine | $24.28 \pm 3.19$ |
| BF without affine | $93.18 \pm 0.06$ |
| **BF** | **$93.27 \pm 0.10$** |

### D.4.1 ABLATION STUDY ON BATCHNORM FREEZING

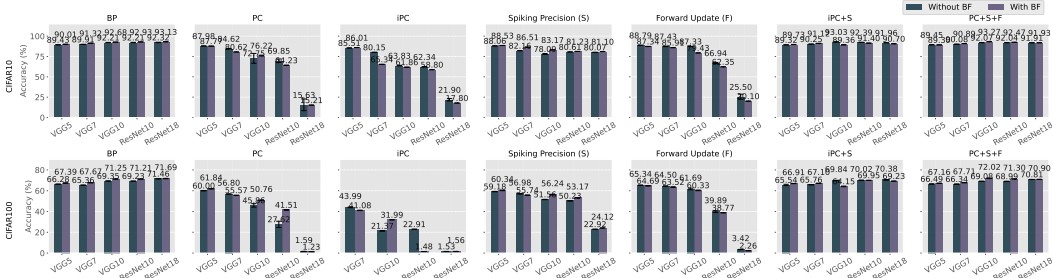

Figure 7: Test accuracies of different algorithms on the CIFAR10/100 datasets across models of varying depths, comparing different methods with and without BatchNorm freezing.

Our investigation reveals that BatchNorm Freezing $(BF)$ significantly enhances model performance when combined with our precision module and forward update mechanism. As illustrated in Figure 7, the integration of $BF$ with our proposed methods $(PC + S + F)$ consistently improved accuracy across all model depths and both CIFAR10 and CIFAR100 datasets. Specifically, the $PC + S + F$ configuration with $BF$ achieved peak performance of 93.27% on CIFAR10, 72.02% on CIFAR100 with the VGG10 architecture and 71.3% on CIFAR100 with the ResNet10 architecture, outperforming even the $BP$ baseline. In contrast, when $BF$ was applied to standard $PC$ or only with forward Update, we observed performance degradation rather than improvement in most cases, the effect of $BF$ was inconsistent and unpredictable across different network depths. These results suggest that the synergy between our proposed components is crucial—$BF$ appears to stabilize the training dynamics specifically when used in conjunction with both our energy balancing mechanisms and forward update. This interaction allows deeper networks to maintain stable gradients throughout the training process, resulting in more robust optimization and ultimately higher classification accuracy.

The effect of BF on *iPC-based models* is more nuanced. While it improves performance on shallower architectures, a slight performance degradation is observed in deeper models. This discrepancy can be attributed to the nature of iPC's update rule. Because iPC updates weights and neural activities simultaneously, updating BN's running statistics requires an *additional, separate weight update step*. This modification alters the computational graph compared to the baseline model (without BF) that was used for the hyperparameter search, which could explain the suboptimal performance in more complex architectures.

## E    HYPERPARAMETER ANALYSIS

### E.1    HYPERPARAMETER IMPORTANCE

To assess the sensitivity of our approach to different hyperparameters, we conducted a hyperparameter importance analysis using a functional ANOVA (fANOVA) based method. This score quantifies the contribution of each hyperparameter to the final optimal validation accuracy. The results for CIFAR10 and CIFAR100 on various architectures are summarized in Tables 17 and 18. The analysis shows that while the learning rate ($lr_w$) is consistently the most critical parameter, our new parameters (like $\alpha/k$ in Precision) are not overly sensitive and show stable influence across settings, indicating a desirable robustness in our proposed methods.

Table 17: Hyperparameter Importance Scores (%) on CIFAR-10.

| Arch. | Method | Activation | T | $\alpha/k$ | $lr_x$ | $lr_w$ | $momentum_x$ | $weight\_decay_w$ |
|-------|--------|-----------|------|-------|-------|-------|-----------|--------------|
| VGG5 | BP | 0.94 | – | – | – | 98.99 | – | 0.07 |
| | PC | 5.96 | 1.00 | – | 77.29 | 10.98 | 3.51 | 1.26 |
| | iPC | 2.20 | 1.15 | – | 1.53 | 88.75 | 1.83 | 3.25 |
| | iPC+S | 0.85 | 0.52 | 0.22 | 1.13 | 94.02 | 0.31 | 2.94 |
| | PC+S+F | 1.68 | 1.63 | 0.53 | 6.64 | 80.44 | 8.19 | 0.89 |
| | PC+D+F | 6.34 | 0.50 | 0.12 | 11.53 | 74.62 | 2.80 | 4.08 |
| VGG7 | BP | 25.94 | – | – | – | 61.56 | – | 12.49 |
| | PC | 15.65 | 0.57 | – | 54.54 | 23.56 | 4.35 | 1.33 |
| | iPC | 1.68 | 1.63 | – | 6.64 | 80.44 | 8.19 | 0.89 |
| | iPC+S | 1.68 | 1.63 | 0.89 | 6.64 | 80.44 | 8.19 | 0.89 |
| | PC+S+F | 3.30 | 0.28 | 4.13 | 10.29 | 54.20 | 1.80 | 25.99 |
| | PC+D+F | 14.78 | 3.05 | 0.01 | 9.00 | 67.45 | 1.17 | 4.54 |
| VGG10 | BP | 9.69 | – | – | – | 87.86 | – | 2.45 |
| | PC | 48.10 | 5.62 | – | 4.02 | 16.15 | 25.32 | 0.79 |
| | iPC | 0.08 | 0.48 | – | 34.55 | 58.50 | 4.73 | 1.66 |
| | iPC+S | 4.85 | 1.80 | 1.72 | 15.15 | 67.61 | 5.64 | 3.22 |
| | PC+S+F | 4.84 | 1.55 | 2.08 | 9.80 | 77.14 | 2.16 | 2.43 |
| | PC+D+F | 8.25 | 2.79 | 0.04 | 1.74 | 75.95 | 9.01 | 2.22 |
| ResNet10 | BP | 6.60 | – | – | – | 93.30 | – | 0.10 |
| | PC | 8.87 | 0.46 | – | 30.41 | 33.60 | 21.77 | 4.89 |
| | iPC | 1.47 | 1.31 | – | 19.18 | 61.87 | 2.87 | 13.30 |
| | iPC+S | 4.74 | 0.22 | 1.07 | 11.68 | 59.33 | 11.79 | 11.16 |
| | PC+S+F | 7.86 | 2.04 | 17.93 | 9.63 | 57.00 | 4.32 | 1.22 |
| ResNet18 | BP | 7.80 | – | – | – | 92.00 | – | 0.20 |
| | PC | 11.39 | 2.86 | – | 55.11 | 12.75 | 8.48 | 9.40 |
| | iPC | 3.76 | 1.57 | – | 14.85 | 47.67 | 11.08 | 21.07 |
| | iPC+S | 5.82 | 0.55 | 1.95 | 2.95 | 78.29 | 8.83 | 1.61 |
| | PC+S+F | 1.73 | 17.97 | 5.98 | 16.95 | 48.95 | 3.05 | 5.36 |

### E.2    HYPERPARAMETER TRANSFERABILITY

To evaluate if hyperparameter searching is required for each setting, we conducted two sets of experiments to evaluate hyperparameter transferability.

**Across Datasets:** We took the optimal hyperparameters found on CIFAR-10 and applied them to CIFAR-100, and vice versa, for the VGG7 architecture.

Table 18: Hyperparameter Importance Scores (%) on CIFAR-100.

| Arch. | Method | Activation | T | $\alpha/k$ | $lr_x$ | $lr_w$ | $momentum_x$ | $weight\_decay_w$ |
|---|---|---|---|---|---|---|---|---|
| VGG5 | BP | 9.49 | – | – | – | 90.50 | – | 0.01 |
| | PC | 3.10 | 3.03 | – | 83.65 | 6.56 | 2.55 | 1.12 |
| | iPC | 1.14 | 2.18 | – | 1.37 | 91.29 | 3.04 | 0.99 |
| | iPC+S | 3.37 | 0.62 | 5.74 | 2.92 | 86.36 | 0.79 | 0.20 |
| | PC+S+F | 7.12 | 8.64 | 1.09 | 1.85 | 75.10 | 5.47 | 0.73 |
| | PC+D+F | 10.43 | 4.03 | 0.07 | 29.70 | 52.92 | 1.50 | 1.35 |
| VGG7 | BP | 3.73 | – | – | – | 92.30 | – | 3.97 |
| | PC | 9.12 | 1.62 | – | 68.17 | 10.75 | 7.14 | 3.20 |
| | iPC | 3.26 | 0.64 | – | 1.87 | 57.30 | 36.54 | 0.40 |
| | iPC+S | 13.47 | 0.37 | 5.47 | 9.96 | 61.09 | 4.30 | 5.34 |
| | PC+S+F | 2.38 | 5.03 | 0.52 | 7.41 | 79.25 | 3.43 | 1.97 |
| | PC+D+F | 6.98 | 0.49 | 0.03 | 0.43 | 86.89 | 0.02 | 5.17 |
| VGG10 | BP | 1.88 | – | – | – | 98.10 | – | 0.02 |
| | PC | 22.79 | 3.60 | – | 6.17 | 62.83 | 3.88 | 0.74 |
| | iPC | 6.01 | 2.51 | – | 4.48 | 40.86 | 43.68 | 2.47 |
| | iPC+S | 4.12 | 0.04 | 6.89 | 5.75 | 77.72 | 3.99 | 1.50 |
| | PC+S+F | 3.76 | 0.40 | 1.25 | 6.71 | 82.10 | 3.04 | 2.74 |
| | PC+D+F | 1.34 | 0.39 | 1.70 | 4.41 | 79.06 | 3.24 | 9.87 |
| ResNet10 | BP | 8.67 | – | – | – | 90.23 | – | 1.10 |
| | PC | 20.05 | 4.46 | – | 10.44 | 46.35 | 14.47 | 4.23 |
| | iPC | 11.93 | 3.77 | – | 9.98 | 59.41 | 13.44 | 1.47 |
| | iPC+S | 0.32 | 2.10 | 6.33 | 21.00 | 44.40 | 1.57 | 24.28 |
| | S+F | 1.93 | 0.53 | 1.23 | 23.06 | 66.37 | 5.77 | 1.12 |
| ResNet18 | BP | 1.49 | – | – | – | 98.21 | – | 0.30 |
| | PC | 0.84 | 4.17 | – | 24.00 | 42.24 | 7.79 | 20.96 |
| | iPC | 2.37 | 7.89 | – | 13.94 | 19.73 | 9.87 | 46.20 |
| | iPC+S | 1.20 | 0.42 | 0.78 | 52.00 | 41.70 | 1.90 | 1.99 |
| | S+F | 1.30 | 0.46 | 0.79 | 14.49 | 34.98 | 45.45 | 2.54 |

**Across Architectures:** We took the optimal hyperparameters from VGG5 and VGG7 and applied them to the VGG10 model on CIFAR-10. Since the hyperparameter T needs to be larger than the number of layers, the T we used in these experiments is $max(10, T_{optimal})$.

The results, presented in Table 19 and 20, suggest that while optimal performance requires dedicated tuning, the hyperparameters show a reasonable degree of transferability, especially for our proposed methods. This indicates they are not pathologically sensitive to the specific dataset or architecture.

Table 19: Test Accuracies from Hyperparameters (HPs) Transfer Across Datasets on VGG7.

| Method | On CIFAR-10 | | On CIFAR-100 | |
|---|---|---|---|---|
| | Optimal CIFAR-10 HPs | Optimal CIFAR-100 HPs | Optimal CIFAR-100 HPs | Optimal CIFAR-10 HPs |
| BP | $89.91^{\pm 0.12}$ | $88.75^{\pm 0.07}$ | $65.36^{\pm 0.15}$ | $65.23^{\pm 0.24}$ |
| PC | $84.62^{\pm 0.10}$ | $78.38^{\pm 0.17}$ | $56.80^{\pm 0.14}$ | $51.65^{\pm 0.28}$ |
| S+F | $90.08^{\pm 0.21}$ | $89.54^{\pm 0.09}$ | $66.34^{\pm 0.22}$ | $63.66^{\pm 0.06}$ |
| iPC | $80.15^{\pm 0.18}$ | $74.35^{\pm 0.90}$ | $43.99^{\pm 0.30}$ | $37.26^{\pm 0.07}$ |
| iPC+S | $90.25^{\pm 0.06}$ | $90.14^{\pm 0.20}$ | $65.76^{\pm 0.12}$ | $61.56^{\pm 1.56}$ |

Table 20: Test Accuracies from Hyperparameters (HPs) Transfer Across Architectures (on CIFAR-10).

| Method | VGG10 with Optimal HPs | VGG10 with VGG7 HPs | VGG10 with VGG5 HPs |
|---|---|---|---|
| BP | $92.21^{\pm 0.08}$ | $90.70^{\pm 0.14}$ | $90.67^{\pm 0.22}$ |
| PC | $72.75^{\pm 6.03}$ | $79.20^{\pm 0.19}$ | $49.66^{\pm 0.54}$ |
| S+F | $92.07^{\pm 0.10}$ | $88.95^{\pm 0.18}$ | $90.67^{\pm 0.11}$ |
| iPC | $63.83^{\pm 0.33}$ | $72.30^{\pm 0.55}$ | $73.78^{\pm 0.15}$ |
| iPC+S | $93.03^{\pm 0.18}$ | $90.60^{\pm 0.60}$ | $92.29^{\pm 0.13}$ |

