# OpenReview forum: "Towards the Training of Deeper Predictive Coding Neural Networks"
_ICLR.cc/2026/Conference — Submitted to ICLR 2026_

### Official Review · Reviewer_vDnV · 2025-10-18

**Soundness:** 3
**Presentation:** 4
**Contribution:** 4
**Rating:** 6
**Confidence:** 4

**Summary:**

The paper tackles why predictive-coding networks (PCNs) fall apart beyond ~5–7 layers and pinpoints the cause as an exponential energy/error imbalance across layers (early layers starved of signal), with an extra timing issue in residual nets where skip-path energy outruns the main path. To fix this, the authors introduce time- and depth-dependent precision schedules—a “spiking precision” that briefly boosts precision when energy first reaches a layer, and a decaying precision that penalizes later layers—plus a forward-update (FU) weight rule that blends initial feed-forward predictions with converged activities to curb deep-layer drift, and BatchNorm Freezing (BF) to stabilize stats during iterative inference; for ResNets they add auxiliary units on residual paths to slow skip energy. Across CIFAR-10/100 and Tiny-ImageNet on VGG/ResNet families, combinations like spiking+FU(+BF) prevent the depth-related accuracy collapse and often match backprop: e.g., VGG-10 hits 93.27% on CIFAR-10 and 72.02% on CIFAR-100 with BF, and VGG-15 on Tiny-ImageNet reaches BP-level performance when spiking+FU+BF are combined. The core takeaway is that precision scheduling + FU re-balances layer energy so deep PCNs can train competitively with BP, albeit with some compute overhead and a slight hit to bio-plausibility from storing forward states.

**Strengths:**

1. Spiking precision is easy to implement and preserves locality. Moreover, iPC+spiking reaches BP-like accuracy on deep CNNs.
2. Identifies exponentially imbalanced layer errors as the root cause, much like vanishing gradients.

**Weaknesses:**

1. The experiments are on small scale datasets only, the reviewer is wondering how it would perform on large scale datasets such as ImageNet.
2. PC/iPC epochs are scale with T×L and possibly slower in training. The reviewer is curious about the setting of T vs depth regrading computational and time cost.
3. The architecture used are all small models. The reviewer is worried about the feasibility of PC/iPC on large, complex networks such as transformer, ResNet101 etc.

**Questions:**

Please see weakness above.

---

> ### Author Response · Authors · 2025-11-20
>
> We thank the reviewer for its time and suggestions. We will address the points one by one:
>
> >  The experiments are on small scale datasets only, the reviewer is wondering how it would perform on large scale datasets such as ImageNet.
>
> We agree that the presented tasks are considered small scale in the current machine learning world, where backprop and GPUs dominate. However, they are still quite challenging when using learning algorithms that are constrained by local computations and other properties that make them interesting for some kind of hardware implementations. However, to test the performance of our method on a larger dataset, we have trained a VGG15 model on a 32×32 downsampled version of ImageNet-1k. For practical reasons, we have used the same hyperparameter as in the Tiny-ImageNet experiments. The results are shown below:
>
> | Method    | Top-1 Accuracy (%) | Top-5 Accuracy (%) |
> |-----------|-------------------|-------------------|
> | BP        | 37.21             | 61.56             |
> | PC+S+F    | 36.09             | 60.92             |
> | iPC+S   | 36.23             | 61.52             |
>
> These results indicate that our method can scale to large datasets with performance close to those of backprop. This suggests that the proposed PC algorithms are effective on large-scale data distributions. Note that we utilized downsampled images as training on full-resolution ImageNet is computationally prohibitive in our current simulation setup, despite it being built in JAX. As discussed in Appendix B, PC introduces computational overhead proportional to the inference steps $T$ (scaling with $T \times L$). Additionally, current GPU libraries do not support parallel updates of neural activities across layers, forcing our simulation to execute sequentially.
>
> Scaling to full resolution would significantly increase the computational cost per trial, making the necessary hyperparameter search infeasible. We acknowledge that experiments on full-resolution ImageNet are an important direction for future work, however, we believe the ImageNet-32 results provide sufficient evidence of the method's feasibility on complex classification tasks.

---

> > ### Author Response · Authors · 2025-11-20
> >
> > > PC/iPC epochs are scale with T×L and possibly slower in training. The reviewer is curious about the setting of T vs depth regrading computational and time cost.
> >
> > As the reviewer correctly inferred, the number of inference steps T must scale with the network depth L to ensure prediction errors can effectively propagate from the output layer back to the input layer. In our practice, to guarantee convergence, the optimal T is usually set within the range [L,1.5L]. However, if it were not for the implementation bottleneck mentioned above, this would not be problematic, as it would have a speed comparable to that backprop. In fact, if we consider ideal parallel hardware (such as neuromorphic chips or optimized parallel GPU kernels):
> > BP Latency: BP has an intrinsic sequential dependency (forward pass followed by backward pass). The error signal must pass through the entire network sequentially, resulting in a theoretical latency of O(L).
> > PC Latency: Under parallel updates, PC latency is determined by the convergence steps T. Since we set T=$[L, 1.5L]$, the theoretical training latency of PC is on the same order of magnitude as BP.
> > To summarize, the current "slowness" arises because we are using serial computation to simulate a process capable of parallel computation. Therefore, while current experiments show slower training for deeper networks, this reflects a simulation bottleneck rather than a lack of viability on the large-scale parallel hardware. We are happy to provide extra details on this if the reviewer is interested, that depends on a problem on applying ‘vmap’ on models with layers of different kinds/dimensions.
> >
> > > The architecture used are all small models. The reviewer is worried about the feasibility of PC/iPC on large, complex networks such as transformer, ResNet101 etc.
> >
> > 1) Resnet101:
> >
> > To validate the feasibility of our proposed methods on deep networks, while mitigating the massive computation and time cost for large-scale ConvNets because of the limitation by the hardware we discussed above. We conduct a experiment using a 101-layer Residual MLP(ResMLP) on MNIST. This proves the effectiveness of our approach also in the case of extremely deep networks, although by using a small scale dataset.
> >
> > Crucially, to test the robustness and ease of tuning of our approach (While also doing such hyperparameter search on a deep model is time assuming), we did not perform a hyperparameter search on the 101-layer model directly. Instead, we adopted a hyperparameter transfer strategy [2]:
> > We performed a hyperparameter search on a 8-layer ResMLP
> > We transferred the lr_w to the 101-layers ResMLP following Depth-\muP scaling rules:
> > 		$$lr_{w}^{(101)} = lr_{w}^{(8)} \times \sqrt{\frac{8}{101}}$$
> > We kept the neural learning rate ($lr_x$) constant and set the inference steps $T = L = 101$.
> > The results are presented below:
> >
> > | Method    | Accuracy (%) |
> > |-----------|--------------|
> > | BP        | 98.40        |
> > | PC+S+F    | 97.13        |
> > | iPC+S+F   | 97.09        |
> >
> > As shown in the table, both PC+S+F and iPC+S+F achieve results comparable to BP, even at a depth of 101 layers. This result is significant given that we use hyperparameters transferred from an 8-layer ResMLP. Considering that PC/iPC have a larger hyperparameter space than BP(due to $T$ and $lr_x$), these results show that our method remains feasible and robust for large-scale deep models.
> >
> >
> > 2) Transformers:
> >
> > As for the transformer, there is very little work on transformer models trained with similar classes of algorithms, as they are quite hard to model/train, and seem to be much more sensitive to hyperparameter changes than their counterpart on GPUs. Furthermore, the general opinion of the community was that it would be almost meaningless to try to scale up predictive coding to transformer models before we manage to do so on ResNets. This is why we believe our work to be important: it addresses this last point, and hence opens up new research towards neuromorphic implementation of transformer models.
> >
> >
> > -----------------------------
> > Please let us know whether we have addressed your concerns, or there are outstanding issues you would still like us to address.

---

> ### Comment · Reviewer_vDnV · 2025-11-21
>
> W1: It looks like the results are not so impressive, which proves my concern that PC/iPC could not perform so well on complex datasets(worse than BP). Nowadays, most BP-free methods could achieve the same accuracy and even beat BP[1-3]. Thus, it weakened the effectiveness of the PC/iPC method. If you could provide me with significant advantages(memory or training time, perhaps) that PC/iPC has over BP, it would be very helpful.
>
> [1] Kappel, David, Khaleelulla Khan Nazeer, Cabrel Teguemne Fokam, Christian Mayr, and Anand Subramoney. "A variational framework for local learning with probabilistic latent representations." In 5th Workshop on practical ML for limited/low resource settings.
>
> [2] Zhang, Aozhong, Zi Yang, Naigang Wang, Yingyong Qi, Jack Xin, Xin Li, and Penghang Yin. "Comq: A backpropagation-free algorithm for post-training quantization." IEEE Access (2025).
>
> [3] Cheng, Anzhe, Heng Ping, Zhenkun Wang, Xiongye Xiao, Chenzhong Yin, Shahin Nazarian, Mingxi Cheng, and Paul Bogdan. "Unlocking deep learning: A BP-free approach for parallel block-wise training of neural networks." In ICASSP 2024-2024 IEEE International Conference on Acoustics, Speech and Signal Processing (ICASSP), pp. 4235-4239. IEEE, 2024.
>
> W2:  The authors' response still does not resolve my concerns for this. If I have understood this correctly, as the training may become slower in practice, the claim that it could be faster is ideal. Could you please provide some concrete evidence to prove that the slow speed is caused by a simulation bottleneck rather than a lack of viability on the large-scale parallel hardware?
>
> W3: I am not sure which community you are referring to. If you mean in the field, with limited hardware access/GPU resources, I agree there's no need to scale up to the transformer. However, if the training time is slower than BP, and the accuracy also cannot surpass BP, I don't think this method would be of sufficient value. Additionally, could the authors compare the trainable parameters with those of BP (CIFAR would be sufficient)?

---

> ### Author Response · Authors · 2025-11-21
>
> We thank the reviewer for the quick answer!
>
> > I am not sure which community you are referring to
>
> As it is stated in the first paragraph of the introduction, we refer to the field that studies learning algorithms for physical/analog hardware/neuromorphic chips, such as memristor crossbar arrays. We study PC not to replace backprop on GPUs, but to enable learning on emerging low-power neuromorphic hardware where backprop is fundamentally infeasible. So, the motivation is that PC or similar algorithms can be used in the regimes where backprop cannot run: in-memory and analog computing, some kinds of neuromorphic systems, and physical neural networks.
>
>
> >  If you could provide me with significant advantages(memory or training time, perhaps) that PC/iPC has over BP, it would be very helpful
>
>
> **Analog Computing**: The significant advantages arise when training actual analog machines/neuromorphic chips, such as memristor crossbar arrays [1].  In fact, the major motivation for PC/Equilibrium Propagation is hardware compatibility with future analog and neuromorphic systems [2]. Unlike BP, equilibrium-style learning uses local updates, does not require storing activations or backpropagating global gradients, and leverages the physical relaxation dynamics of the hardware itself. Thus, while the benchmarks we propose in this work mainly reflect digital performance, PC/Eqprop offers a path toward high-efficiency training on emerging neuromorphic hardware, where we expect its advantages to become much more significant.
>
> [1] Yi, Su-in, et al. "Activity-difference training of deep neural networks using memristor crossbars." Nature Electronics 6.1 (2023): 45-51.
>
> [2] Scellier, Benjamin, et al. "Energy-based learning algorithms for analog computing: a comparative study." Advances in neural information processing systems 36 (2023): 52705-52731.
>
> **Efficiency**: The current implementation is a GPU simulation of dynamics that ideally will be then performed directly on hardware, where computations occur natively in parallel through the physics of the analog substrate. In such systems, settling to equilibrium happens without iterative digital computation, eliminating the main source of runtime overhead observed in simulation. Therefore, while the GPU implementation is slower, this reflects a software approximation rather than the efficiency of the actual hardware learning algorithm, where we expect significantly faster training.

---

> ### Author Response · Authors · 2025-11-21
>
> > Nowadays, most BP-free methods could achieve the same accuracy and even beat BP[1-3]
>
> We appreciate the reviewer’s point, but we would like to clarify that BP-free is not a single class of methods that can be compared against each other, as different approaches impose very different architectural and computational constraints that depend on their intended application: Methods that aim for full BP-level performance on distributed GPUs often rely on additional mechanisms (some of the ones you cited) that are not compatible with analog or low-power hardware. In contrast, more constrained approaches such as equilibrium propagation and predictive-coding-style updates are explicitly designed for local learning and physical substrates, and such constraints naturally hurt performance. There also exist BP-free methods with even stronger hardware constraints than PC/Eqprop that currently perform far worse in terms of test accuracies. For these reasons, we do not believe that comparing BP-free methods among each other based only on test accuracies, without considering their characteristics, is the right way of judging them.
>
> **Example**. There is a very influential method that is used in analog hardware, called PEPITA [1]. This algorithm has the potential of being used in photonics [2]. However, in terms of test accuracies in simulations, it performs much worse than PC/Eqprop. However, it would be extremely unfair to state that the algorithm is not important as it reaches worst accuracies than PC/Eqprop.
>
> [1] https://proceedings.mlr.press/v162/dellaferrera22a.html?trk=public_post_comment-text
> [2] https://opg.optica.org/abstract.cfm?uri=OFC-2025-Tu3F.3
>
> **References**: Thank you for the pointers. However, the algorithms proposed by the works you have cited do not have the characteristics needed to be applied directly on the hardware. We went through the papers, and realized that your reference [1] still uses a local form of backprop (applied to individual blocks), and hence would carry the problems of backprop on hardware; [2] is a post-training algorithm, and hence not related to what we propose, while [3] is perhaps the one that shares some similarities. In fact, it is a block-wise local-loss method trained with standard SGD, and the local nature of the updates (each block learns independently without global gradient transport) could facilitate future analog or neuromorphic implementations. However, additional changes would be required to fully align with equilibrium-based learning: the model should replace explicit SGD with local activity-based updates, introduce a physical inference/relaxation phase, and enforce bidirectional constraints so that prediction errors propagate through the system’s natural dynamics. If we add such constraints, their test accuracy would deteriorate.
>
> We believe that, if we remove the hardware constraints from PC/Eqprop, we would be able to reach competitive performance to the works you are citing. However, this would invalidate the goal of this field of research. To conclude, we'd like to state that we are confident about the results we propose: given the constraints we consider in our class of algorithms, the performance we report in this work are state of the art in the field, as you can easily check against the two recent highly cited works that study this summarize the performance of this class of algorithms [1,2].
>
> [1] Pinchetti et al. Benchmarking Predictive Coding Networks -- Made Simple:  https://openreview.net/forum?id=sahQq2sH5x
>
> [2] Scellier, Benjamin, et al. "Energy-based learning algorithms for analog computing: a comparative study." Advances in neural information processing systems 36 (2023): 52705-52731.
>
>
> > W2: The authors' response still does not resolve my concerns for this.  Could you please provide some concrete evidence to prove that the slow speed is caused by a simulation bottleneck?
>
> This is discussed in the work that presents the library we are using for the simulations: https://openreview.net/forum?id=sahQq2sH5x .
>
> In Page 9, the authors state (copy-paste):
>
> **Limitations.**  The efficiency of PCX could be further increased by fully parallelizing all the operations. In fact, in its current state, JIT is unable to parallelize the execution of the layers; a problem
> that can be addressed with the JAX primitive vmap, but only in the unpractical case where all the
> layers have the same dimension. To test how different hyperparameters of the model influence the
> training speed, we have taken a feedforward model, and trained it multiple times, each time increasing
> a specific hyperparameter by a multiplicative factor. The results, reported in Fig. 8, show that the
> two parameters that increase the training time are the number of layers L and the number of steps T .
> Ideally, only T should affect the training time as inference is an inherently sequential process that
> cannot be parallelized, but this is not the case, as the time scales linearly with the amount of layers.

---

### Official Review · Reviewer_pzZZ · 2025-10-30

**Soundness:** 3
**Presentation:** 3
**Contribution:** 2
**Rating:** 4
**Confidence:** 2

**Summary:**

Based on the observation that predictive coding networks suffer significant performance degradation beyond five to seven layers despite are effective in shallow architectures, this work proposes two algorithmic improvements (spiking precision and a novel weight-update mechanism) and two structural improvements (PCtailored batch normalization and auxiliary neurons for skip connections) that enable PC to achieve competitive performance with backprop on image classification benchmarks. While the motivation is clear, the related background and contribution significance should be clarified in detail.

**Strengths:**

This work may first reveal that the energy is orders of magnitude larger in layers closer to the output  in models trained with predictive coding (PC).

To regulate the energy imbalance and improves test accuracy in deep PC models and the case  incremental PC (iPC), this work proposes dynamical precision-weightings that depend on both time and layer depth, e.g. spiking precisions, that can achieve performance comparable to backpropagation in deep networks.

To achieve the goal of slowing down the feedback signal of the skip connections so that it reaches the higher layers at the same time as the main one, this work proposes to add extra families of neurons inside the skip connection that can reach performance comparable to these of backprop on ResNet18 with PC and iPC.

**Weaknesses:**

More details are needed to derive Equations (2) and (3).

Are the model and analysis suitable for transformer architecture?

If the algorithm's effect is only comparable to the backpropagation effect, what is its significance or advantage? It is necessary to first explain the relevant background or significance in detail.

The comparison of relevant algorithms still needs to be strengthened, and the contribution significance over related state-of-the-art works should be highlighted.

A pseudocode needs to be added to demonstrate how this algorithm operates and how to conduct related experiments.

**Questions:**

Please see the Weaknesses.

---

> ### Author Response · Authors · 2025-11-20
> **Rebuttal**
>
> We thank the reviewer for its time and suggestions. We will first focus on your main concern: what is the significance or advantage of studying algorithms like equilibrium propagation and predictive coding.
>
> > If the algorithm's effect is only comparable to the backpropagation effect, what is its significance or advantage?
>
> We study PC/Equilibrium Propagation models not to replace backprop on GPUs, but to study a class of algorithms that could enable learning on emerging low-power neuromorphic hardware where backprop is fundamentally infeasible. So, the motivation is that PC or similar algorithms can be used  in the regimes where backprop cannot run — on-device, energy-constrained, neuromorphic systems, and physical neural networks.
>
> > It is necessary to first explain the relevant background or significance in detail.
>
> We have explained it as follows at the beginning of the paper (in the first paragraph of the introduction):
>
> *<<Training deep learning models is extremely expensive in terms of energy consumption. To address this problem, a recent direction of research is studying the use of alternative accelerators that leverage the properties of physical systems to perform computations, such as in-memory computations using memristor crossbars. However, transitioning to new hardware without altering the main algorithm — error backpropagation — has proven to be challenging due to two central issues: the requirement of sequential forward and backward passes, and the need to analytically compute gradients of a global cost function. These issues do not arise when using learning algorithms that rely on computations that are local in space and time.>>*
>
>  However, we have updated the manuscript to make it clearer, iterating this again at a later stage to make the concept clearer.
>
> > Are the model and analysis suitable for transformer architecture?
>
> There is very little work on transformer models trained with similar classes of algorithms, as they are quite hard to model/train, and seem to be much more sensitive to hyperparameter changes than their counterpart on GPUs (one exception for a 2-layer model: https://proceedings.neurips.cc/paper_files/paper/2022/hash/08f9de0232c0b485110237f6e6cf88f1-Abstract-Conference.html). Furthermore, the general opinion of the community was that it would be almost meaningless to try to scale up predictive coding to transformer models before we manage to do so on ResNets. This is why we believe our work to be important: it addresses this last point, and hence opens up new research towards neuromorphic implementation of transformer models.
>
> > The comparison of relevant algorithms still needs to be strengthened, and the contribution significance over related state-of-the-art works should be highlighted.
>
> We believe we have discussed this properly in the related works and in the introduction. More in detail, in the second paragraph of the introduction, we cite and discuss the two main papers that report state of the art results in the field of NeuroAI/neuromorphic learning algorithms. In the related works, we have discussed/cited a large amount of works that use predictive coding/equilibrium propagation to perform  supervised learning.
>
> However, if you believe there are other works we have not cited/discussed that should be present to strengthen the contribution of the paper, please let us know which ones and we would of course be happy to add such discussions either in the main body or in the supplementary material.
>
> About state of the art results, our work is the first to extend predictive coding to very deep convolutional networks (VGG15) and residual networks (ResNet18) on complex benchmarks such as CIFAR-100 and TinyImageNet, while achieving performance comparable to backpropagation, hence solving the open problem proposed by Pinchetti et al. 2024.
>
> Other concerns:
>
> > More details are needed to derive Equations (2) and (3).
>
> Equations 2 and 3 are gradient descent updates on the energy function of Equation (1), where we have considered $\mu^{l}_t = \mathbf{W}^{l} f\left( \mathbf{x}^{l-1}_t \right)$. Hence, they are computed by computing the derivative of Equation (1) under **x** and under **W**, respectively. We have stated this more clearly in the background section.
>
> > A pseudocode needs to be added to demonstrate how this algorithm operates and how to conduct related experiments.
>
> We agree that pseudocode improves reproducibility and clarity. We have added the pseudocode to the Appendix in the revised version.
>
> -----------
> -----------------------------------------
>
> Conclusion:
>
> May we ask whether the main reason for your “rejection score” was due to the belief that we were proposing to replace BP with PC for standard tasks with GPUs? If that was the case, have we properly addressed your primary concern?
>
> -------------------------------------------

---

> > ### Comment · Reviewer_pzZZ · 2025-11-26
> > **I will maintain my original score.**
> >
> > Thanks for the rebuttal. However, the authors still do not convince me about the applicability of the predictive coding. Though it achieves  performance comparable to backpropagation with VGG15 or ResNet18 on CIFAR-100,TinyImageNet, the authors do not show whether this comparable performance is generalized to the more popular architecture, i.e. Transformer (this point is also pointed out by the other reviewers) on more challenging dataset.  Meanwhile, comparing with only 'two main papers' is not sufficient for me to draw conclusions about the innovation or the significance of the contribution of this paper. Thus, I will maintain my original score.

---

> > > ### Author Response · Authors · 2025-11-26
> > >
> > > Dear Reviewer,
> > >
> > > Thank you for your reply. However, we respectfully disagree with the underlying philosophy of your judgment, especially regarding transformer architectures and the applicability of predictive coding.
> > >
> > > **Transformers**:
> > > It is accurate that we do not include transformer results. But no existing work under the same analog/physical constraints does either. If papers in this field are judged solely on whether they already scale to transformers, then meaningful research progress becomes impossible. Fortunately, this is not the community consensus: ~40 PC/EqProp papers have been accepted at NeurIPS/ICLR/ICML in the past two years (including spotlights/orals), and none show competitive transformer results beyond tiny toy models.
> > >
> > > We fully understand your point: demonstrating scalability to transformers would be a major breakthrough. But your criticism does not seem specific to our contribution; it reads as a blanket statement that “this field is not worth publishing until it reaches transformers.” If every intermediate step is rejected on that basis, how can we ever get there?
> > >
> > > **Applicability of Predictive Coding**:
> > > Predictive coding may not be as “trendy” or immediately industrial as transformer-based deep learning, but ICLR explicitly welcomes work in NeuroAI, bio-plausible learning, and neuromorphic computing. We believe that reviewers should judge contributions within the context of their field, not dismiss an entire research direction a priori.
> > >
> > > **Baselines / References**:
> > > The two papers we compare to are benchmark works that summarize the current state of the art in predictive coding. All prior results are either included within them or significantly weaker. So we are not just improving on two papers, we are improving over every relevant work from the last ~5 years (all in top-tier ML venues).
> > >
> > > We find it unfair to state that the references are “unconvincing” without specifying what is missing. If there truly are works that contradict our claim of SOTA performance, please point us to them: we would be happy to discuss this.
> > >
> > > ⸻
> > >
> > > In summary, we have three concrete questions:
> > >
> > > 1)  Is there any predictive-coding work that achieves larger-scale architectures or higher performance than ours? If so, which one?
> > > 2) If no method in this field currently scales to transformers, do you think this field should not exist?
> > > 3)  What reference or baseline do you believe we should include that would change your opinion?

---

### Official Review · Reviewer_MFEU · 2025-11-01

**Soundness:** 2
**Presentation:** 1
**Contribution:** 2
**Rating:** 2
**Confidence:** 4

**Summary:**

The paper diagnoses the main failure mode of deep predictive-coding networks as a severe imbalance of prediction-error energy across layers, whereby upper layers accumulate large errors while lower layers receive effectively vanishing signals. To address this, it introduces time- and depth-dependent precision schedules, a forward-aware weight update, and architectural adjustments (e.g., BN freezing, auxiliary units on skip paths) that redistribute error energy and enable PC/iPC to approach backprop-level performance on deeper VGGs and ResNet-18 on Tiny-ImageNet.

**Strengths:**

It proposes practical, PC-compatible remedies—precision scheduling, forward-aware updates, and small architectural tweaks—that are simple to implement yet demonstrably restore performance to near–backprop levels on deeper CNNs, including ResNet-18 on Tiny-ImageNet.

**Weaknesses:**

**1. Framing/positioning issues**

* Even though energy-based models (EBMs) in deep learning have continuously advanced (e.g., JEM, diffusion models), the authors single out only Hopfield-style energy functions and predictive coding as EBMs and frame the work as an effort to “scale up” a very general EBM framework; this characterization is somewhat misleading.

**2. Notation / mathematical clarity**

* Notation and formatting are inconsistent. At line 146, the function $f$ is used without prior definition, and it is typeset in bold in one instance and non-bold in another. Similarly, several vectors are sometimes bolded and sometimes not; $\mathbf{y} \in \mathbb{R}^o$ (line 185) reuses $o$ that later seems to denote an input $\mathbf{o}$, creating an avoidable clash; and the expression “learning rate” is used for both inference- and learning-phase coefficients, which is imprecise. These issues make the mathematical development harder to follow than necessary.

* In the background section, “covariance of a specific neuron” is described, but covariance usually is defined over a set/list of variables, not a single unit; moreover, the text does not consistently distinguish between variance and covariance, which obscures what is actually being estimated and updated.

* Equation (3) appears to concatenate $\boldsymbol{\epsilon}$ and $f(\mathbf{x})$, both column vectors, in a way that is not well defined as written. The base model in the background section omits bias terms without explicitly stating this assumption, together with sporadic typos (e.g., “$x^0_0$” instead of “$\mathbf{x}^0_0$” at line 187, which weakens the technical polish of the paper.

**3. Figures/presentation**

* Figures are not reader-friendly. In Fig. 2, the train/val accuracy is difficult to distinguish by color, and the multiple per-layer curves seem to be intended to demonstrate energy imbalance; they are visually crowded. If layerwise imbalance is a key claim, a more quantitative or tabular display would present it more clearly. Similar color/legibility issues recur in other figures.

**4. Structure/organization**

* The organization is somewhat scattered: observations about energy imbalance appear inside the method section together with experimental results, and are then revisited in a separate section, which blurs the line between “what the method is” and “what we observed.” A more precise separation of motivation → method → experiments would improve readability.

**5. Novelty/relation to prior work (https://openreview.net/pdf?id=s3E08R4AMK)**

* Conceptually, much of the contribution overlaps with prior work that has already identified layerwise energy/error imbalance in PC. The proposed solution—time/depth precision spikes and forward updates—also looks very close to earlier versions of the work, raising questions about the level of novelty and, in particular, why a spike-shaped schedule is the right or unique choice.

**6. Theoretical grounding/justification**

* The path from the diagnosed problem to the proposed solution feels somewhat ad hoc. Beyond empirical evidence that the schedule “works,” the paper does not provide a stronger theoretical rationale (e.g., analysis of convergence or of error-energy propagation under the proposed precision) to justify why this particular scheduling and covariance treatment is appropriate.

**7. Methodological detail/completeness**

* Several key methodological elements are under-explained: it is not clearly specified *what* the covariance is taken over, *how* samples for covariance are collected during PC inference, *how* the substantial computational cost of covariance estimation is managed, and *why* temporal scheduling is necessary on top of the basic covariance idea. In addition, Eq. (2) already introduces (\alpha), but Eq. (4) seems to reuse (\alpha) inside (\Sigma); it is unclear whether this is intentional scaling or accidental duplication, and the resulting (\Sigma) no longer looks like a covariance. This also makes the method look very close to existing work.

**Questions:**

1. In an earlier version of this line of work, VGG-13 was included, but it is omitted here; is there a specific reason for excluding networks deeper than 10 layers, and does the proposed precision/covariance scheme still work reliably when depth exceeds 10?

2. Can the authors clarify the exact definition and estimation procedure for the covariance term (over which variables, over what time/window, and with what computational budget), and whether a lower-cost approximation was considered?

3. Given that several recent papers have already reported layerwise energy/credit imbalance in predictive coding, can the authors more precisely delineate what is new here beyond revalidating that observation—especially regarding the choice of a spike-shaped schedule?

---

### Official Review · Reviewer_BswA · 2025-11-01

**Soundness:** 1
**Presentation:** 1
**Contribution:** 2
**Rating:** 2
**Confidence:** 3

**Summary:**

The manuscript presents a modification to the standard training pipeline for Predictive Coding Networks (PCN). The proposed approach addresses the imbalance in energy distribution between the first and last layers that leads to a performance drop. The key contributions are spiking precision, forward updates, and structural adjustments for skip-connections and batch norm layers. The authors sequentially analyze the impact of the introduced modifications on the energy distribution and the resulting accuracy. The final experiments demonstrate that the proposed techniques preserve PCN accuracy in deeper models.

**Strengths:**

The main strength of the submission is the clear correspondence between the stated problem and the main results (in Table 2). The proposed modifications improve PCN accuracy for deeper models. In addition, the proposed structural update to skip connections and batch normalization improves accuracy for ResNet models. Moreover, the more layers there are, the greater the gain.

**Weaknesses:**

Below, I list the weaknesses observed in the presented submission:
1. The motivation for using PCN instead of the standard backpropagation (BP) remains unclear to me. Experiments demonstrate that the BP typically achieves higher test accuracy; therefore, the rationale for considering such a training scheme must be clearly explained at the outset, highlighting the advantages of PC over BP.
2. Experiments are smoothly distributed over the sections, and many references to Figures are missing in the sections, for example, paragraphs "Results" in sections 4.2, 4.1.1, 4. Therefore, it is hard to determine which data support the results presented in the corresponding paragraphs.
3. I do not see any explanations why the training procedure presented in section 3 corresponds to the classification error minimization. The equations (1)-(3) do not align with the original task to make the best classification model.
4. The proposed modifications are not described with the equations (e.g., Forward Updates and Structural Updates) or lack a clear intuition (I do not say about theoretical derivation) like in the Spiking Precision paragraph. Therefore, fair analysis and comparison with baselines are impossible.
5. I do not see any runtime or memory comparison between the proposed approaches and the mentioned baselines.
6. Center nudging appears to be useless while the spiking mechanism works (Figure 3)

**Questions:**

1. Why can the proposed method be practically important? What use cases are the most suitable for using the presented approaches?
2. I see the comparison only on the classical architectures, while the current best results are obtained with the transformer-like models. Do you have any results for this or similar attention-based architectures?
3. Do you have any ideas about deriving theoretical guarantees on the convergence improvement observed empirically?
4. How robust are the proposed modifications to the noise on the gradient estimate? What is the recommended batch size for the datasets under consideration?
5. Why do the proposed approaches fail for the ResNet18 model and underperform compared to BP?

---

> ### Author Response · Authors · 2025-11-20
> **Rebuttal**
>
> We thank the reviewer for its time and suggestions.
>
> > The motivation for using PCN instead of the standard backpropagation (BP) remains unclear to me. Experiments demonstrate that the BP typically achieves higher test accuracy; therefore, the rationale for considering such a training scheme must be clearly explained at the outset, highlighting the advantages of PC over BP.
>
> We study PC not to replace backprop on GPUs, but to enable learning on emerging low-power neuromorphic hardware where backprop is fundamentally infeasible. So, the motivation is that PC or similar algorithms can be used in the regimes where backprop cannot run: in-memory and analog computing, some kinds of neuromorphic systems, and physical neural networks.
>
> -------------------------------
>
> > The rationale for considering such a training scheme must be clearly explained at the outset, highlighting the advantages of PC over BP.
>
> We believe we do this in the very first paragraph of the introduction, that we quote below:
>
>
> *<<Training deep learning models is extremely expensive in terms of energy consumption. To address this problem, a recent direction of research is studying the use of alternative accelerators that leverage the properties of physical systems to perform computations, such as in-memory computations using memristor crossbars. However, transitioning to new hardware without altering the main algorithm — error backpropagation — has proven to be challenging due to two central issues: the requirement of sequential forward and backward passes, and the need to analytically compute gradients of a global cost function. These issues do not arise when using learning algorithms that rely on computations that are local in space and time.>>*
>
>
> We will rephrase this to make it clearer. This should also address your first question:
>
>
> > Why can the proposed method be practically important? What use cases are the most suitable for using the presented approaches?
>
> To iterate more: On GPUs, we agree with you: there is no evidence on why we would use PC instead of BP. However, this changes when we move away from GPUs towards low-energy neuromorphic chips, such as analog hardware, in-memory computing. In such cases, backprop has been proven to be extremely unstable in some cases, not possible to be implemented in others. The field studies algorithms that would work in setups where BP cannot be used, or is suboptimal, that is in neuromorphic computing.
>
> -------------------------------------------------------------
>
> Here are two comments we do not fully understand, and we would like to have more clarification about:
>
> > Center nudging appears to be useless while the spiking mechanism works (Figure 3)
>
> This is correct: when using spiking precisions, our experimental evidence suggests that centered nudging does not provide the same advantages it provides with standard predictive coding. What we did not understand is why is this considered a weakness of the work? It is simply a property we have unveiled, and transparently reported.
>
> > Why do the proposed approaches fail for the ResNet18 model and underperform compared to BP?
>
>
> Can we ask why you believe that our approach *fails* on ResNet18? In none of the experiments we have proposed we observe a  performance gap larger than 1.5% between backprop and one of the proposed algorithms. We believe this to be a positive result, as this means that we managed to perform almost as well as backprop, by using a learning algorithm that also has additional hardware-aware constraints.
>
> ---------------------------------------------------------------------------
>
> As we have discussed the motivation of our work, may we ask whether the reason for your strong rejection was due to the fact that you thought we were proposing to replace BP with PC for standard tasks with GPUs? If that was the case, have we properly addressed your primary concern?

---

> ### Author Response · Authors · 2025-11-20
>
> > I see the comparison only on the classical architectures, while the current best results are obtained with the transformer-like models. Do you have any results for this or similar attention-based architectures?
>
>
> There is very little work on transformer models in the field of neuromorphic computing, as they are quite hard to model/train, and seem to be much more sensitive to hyperparameter changes than their counterpart on GPUs. Furthermore, the general opinion of the community was that it would be almost meaningless to try to scale up predictive coding to transformer models before we manage to do so on ResNets. This is why we believe our work to be important: it addresses this last point, and hence opens up new research towards neuromorphic implementation of transformer models.
>
>
> > Experiments are smoothly distributed over the sections, and many references to Figures are missing in the sections, for example, paragraphs "Results" in sections 4.2, 4.1.1, 4. Therefore, it is hard to determine which data support the results presented in the corresponding paragraphs.
>
> We apologize for this, but our plan was to structure the paper in:
> 1)  first hypothesis (algorithmic) - empirical evidence that confirms the hypothesis;
> 2)  second hypothesis (structural) - empirical evidence that confirms the hypothesis;
> 3)  a more experimental final section that shows how merging the two methods allows us to address the open problems stated at the beginning of the paper.
>
> We have changed the names of the first subsection in  “Experimental Validation of the Algorithmic Contribution”;  while the main one in “Larger Scale Experiments”. About referencing the figures and tables: we have doublechecked the manuscript, and all the tables and figures are correctly referenced in the right subsection. Is there something we are missing there?
>
> > I do not see any explanations why the training procedure presented in section 3 corresponds to the classification error minimization. The equations (1)-(3) do not align with the original task to make the best classification model.
>
> At initialization, minimizing the predictive coding energy is exactly equivalent to minimizing the supervised loss: after the forward pass, all internal prediction errors are zero, so when the output neurons are clamped to the label, the entire energy corresponds to the output mismatch — i.e., the classification objective. During inference (t>0), prediction errors propagate inward to assign credit locally, as is standard in supervised predictive coding (e.g., Whittington & Bogacz 2017; Song et al. 2020).
>
> We acknowledge that this connection may not have been sufficiently explicit in the manuscript. We have now revised the explanation around lines ~187 to make it clearer.
>
>
> > I do not see any runtime or memory comparison between the proposed approaches and the mentioned baselines.
>
> They are reported in the supplementary material: please see Section B.
>
> > How robust are the proposed modifications to the noise on the gradient estimate? What is the recommended batch size for the datasets under consideration?
>
> Our method makes the weight update to be closer to that of backprop: In standard PC we have approximation results, but we have shown that the magnitude of PC’s updates diverges a lot from that of backprop. Spiking precisions address this. Furthermore,  forward updates make sure that half of the parameters needed for weight updates (in this case, the pre-synaptic neurons), are the same as backprop. About the batch size, you can find a detailed discussion on this and all the other hyperparameters in the supplementary material, where we also compute the importance score of each of them.
>
> > The proposed modifications are not described with the equations
>
> Thank you for the pointer: We have addressed this by better rephrasing the description of the algorithms. Notably, we have added added two new paragraphs that provide a formal description and derivation of the new energy function that emerges when adding auxiliary neurons in the residual connections. We refer to the updated manuscript for further details.

---

### Meta-Review · Area_Chair_nGx6 · 2025-12-08

**Summary:**

The paper presents a set of engineering improvements that substantially stabilize deep predictive coding/iPC on VGG/ResNet-style networks and bring performance close to BP on CIFAR/Tiny-ImageNet–scale tasks. However, the overall contribution remains limited in scope: the methods are largely heuristic, the exposition leaves key technical details opaque, and the work does not clearly demonstrate advantages over standard BP or other BP-free/local-learning approaches at a scale that would be compelling for a broad ML audience. I therefore recommend rejection, while acknowledging its potential value for more specialized PC/NeuroAI venues.

**Reviewer Concerns:**

The rebuttal partially addresses some factual misunderstandings (e.g., the existence of >10-layer experiments and the relationship to the prior workshop version) and clarifies aspects of the experimental setup, which mitigates the strongest “no novelty” and “no deep results” claims. However, important concerns remain largely unresolved: the derivation and transparency of the training rules (precision schedule, forward update), the lack of a clear theoretical rationale beyond heuristics, and the limited experimental scope and impact relative to BP and recent BP-free methods. As a result, I find the core concerns about clarity, soundness (in the sense of reproducibility/interpretability), and significance still outstanding.

**Reviewer Scores:**

For the reviewer(s) whose main objections were based on factual errors about depth and prior work overlap, I expect only a modest upward adjustment (e.g., from clear reject to borderline) after discussion, since their broader concerns about impact and framing would likely persist. For the reviewer(s) emphasizing unclear methodology and limited scale/novelty, I do not anticipate a meaningful score change, as the rebuttal did not substantially alter the strength of the evidence or the clarity of the presentation. Overall, even with minor upward shifts for one reviewer, the consensus would likely remain below the acceptance threshold.

---

### Decision · Program_Chairs · 2026-01-26

Reject